# Activation loop phosphorylation and cGMP saturation of PKG regulate egress of malaria parasites

Konstantinos Koussis[1¤]*, Silvia Haase[2], Chrislaine Withers-Martinez[1], Helen R. Flynn[3], Simone Kunzelmann[4], Evangelos Christodoulou[4], Fairouz Ibrahim[3], Mark Skehel[3], David A. Baker[5], Michael J. Blackman[1,5]*

1 Malaria Biochemistry Laboratory, Francis Crick Institute, London, United Kingdom, 2 Host-Pathogen Interactions in Cryptosporidiosis Laboratory, The Francis Crick Institute, London, United Kingdom, 3 Proteomics Science Technology Platform, The Francis Crick Institute, London, United Kingdom, 4 Structural Biology Science Technology Platform, The Francis Crick Institute, London, United Kingdom, 5 Faculty of Infectious and Tropical Diseases, London School of Hygiene & Tropical Medicine, London, United Kingdom

¤ Current address: Faculty of Infectious and Tropical Diseases, London School of Hygiene & Tropical Medicine, London, United Kingdom
* konstantinos.kousis@lshtm.ac.uk (KK); mike.blackman@crick.ac.uk (MJB)

**Data Availability Statement:** The mass spectrometry proteomics data have been deposited to the ProteomeXchange Consortium via the PRIDE (65) partner repository (http://

## Abstract

The cGMP-dependent protein kinase (PKG) is the sole cGMP sensor in malaria parasites, acting as an essential signalling hub to govern key developmental processes throughout the parasite life cycle. Despite the importance of PKG in the clinically relevant asexual blood stages, many aspects of malarial PKG regulation, including the importance of phosphorylation, remain poorly understood. Here we use genetic and biochemical approaches to show that reduced cGMP binding to cyclic nucleotide binding domain B does not affect *in vitro* kinase activity but prevents parasite egress. Similarly, we show that phosphorylation of a key threonine residue (T695) in the activation loop is dispensable for kinase activity *in vitro* but is essential for *in vivo* PKG function, with loss of T695 phosphorylation leading to aberrant phosphorylation events across the parasite proteome and changes to the substrate specificity of PKG. Our findings indicate that *Plasmodium* PKG is uniquely regulated to transduce signals crucial for malaria parasite development.

## Author summary

Despite all efforts to control and eradicate malaria, the disease still poses a huge burden on human health. Almost half of the world's population lives in high malaria transmission areas with over half a million deaths occurring annually due to the disease, which is caused by a single-celled parasite called Plasmodium. Replication of the parasite inside red blood cells is responsible for all the clinical manifestations of the disease. At the end of each replicative cycle, parasites rupture the red blood cell in a process known as egress in order to invade new red blood cells. Previous studies have shown that an essential kinase termed PKG is a master regulator of egress. However, many aspects of PKG regulation are still

proteomecentral.proteomexchange.org) with the dataset identifiers PXD049083 (P. falciparum T695 phosphorylation IP), PXD048979 (P. falciparum IP) and PXD049017 (P. falciparum TMT Phosphoproteome).

**Funding:** The work was supported through a joint Investigator Award from the Wellcome Trust to MJB (220318/A/20/Z) and DAB (220318/Z/20/Z). This work was also supported by the Francis Crick Institute (https://www.crick.ac.uk/) which receives its core funding from Cancer Research UK (CC2129), the UK Medical Research Council (CC2129), and the Wellcome Trust (CC2129). The work was further supported by Wellcome ISSF2 funding to the London School of Hygiene and Tropical Medicine. The funders had no role in study design, data collection and analysis, decision to publish, or preparation of the manuscript.

**Competing interests:** The authors have declared that no competing interests exist.

unknown. In this work we examined the importance of phosphorylation on PKG function by replacing wild type PKG with mutant forms refractory to phosphorylation. We found that phosphorylation in a specific region of the protein is essential for parasite survival. Excitingly though, phosphorylation is not essential for kinase activity, as has been shown for mammalian PKG proteins but regulates its substrate specificity. Our results suggest that Plasmodium PKG is uniquely regulated compared to its mammalian counterparts, to facilitate parasite proliferation.

## Introduction

Parasitic protozoa encounter diverse environments as they transition between different host organisms and cell types during their life cycle. For *Plasmodium* spp., the causative agents of malaria, proliferation within vertebrate and mosquito vector host cells relies on the timely response to a range of environmental signals. *Plasmodium falciparum* causes the most severe form of malaria in humans, resulting in more than 600,000 deaths in 2022 [1]. Asexual blood stage parasites are responsible for all the clinical manifestations of the disease. At the end of each replicative cycle, merozoites egress from the confines of the RBC in response to a tightly regulated signalling cascade which leads to the protease-dependant destabilisation and rupture of the bounding parasitophorous vacuole and RBC membranes [2,3].

Second messengers and in particular cyclic 3',5'-guanosine monophosphate (cGMP) play an essential role throughout the *Plasmodium* life cycle. Current knowledge suggests that all cGMP signalling in the parasite is mediated through a single cGMP-dependent protein kinase (PKG) [4,5]. Genetic and chemical approaches have shown that *Plasmodium* PKG has essential roles in almost all life cycle stages, including liver stage development and merosome egress, formation of gametes (gametogenesis), ookinete conversion and motility in the mosquito vector as well as sporozoite motility and infection [6–9]. In the asexual blood stages that are responsible for disease, application of selective inhibitors of PKG has shown that kinase activity is dispensable for intraerythrocytic parasite development but essential for the final step of merozoite egress [10–12]. More recently, conditional genetic approaches have confirmed the above findings, additionally showing that PKG has no essential scaffolding or adaptor role during intraerythrocytic development but interacts with a transmembrane protein (ICM1) involved in calcium mobilisation at egress [13–15].

Like all other apicomplexan parasites, malaria parasites encode a single PKG gene with unusual structural properties, including the presence of four cyclic nucleotide domains as compared to two in mammalian PKG counterparts. In the coccidian subgroup of apicomplexan parasites (such as *Toxoplasma* and *Eimeria*), the PKG gene possesses two alternative transcriptional start sites, resulting in the expression of two isoforms of PKG that localise to the cytosol and the plasma membrane, respectively, the latter through myristoylation [16, 17]. This plasma membrane association of PKG is essential for its function in *Toxoplasma* [18]. In contrast, the *Plasmodium* PKG gene lacks both the alternative transcriptional start site and the consensus myristoylation motifs, but a sub-population of PKG molecules are thought to be associated with the ER membrane, possibly through interaction with ICM1 [12, 15, 17]. Structural studies have shown that *Plasmodium* PKG adopts a "pentagonal" multi-domain structure comprising a central kinase catalytic domain surrounded by one degenerate (CNB-C) and three functional (CNB-A, -B and -D) cyclic nucleotide-binding domains [19]. Analysis of recombinant PKG or isolated domains thereof have shown that CNB-D binds cGMP with the highest affinity and selectivity relative to cAMP. Saturation of CNB-D was shown to be a

prerequisite for PKG activation, while mutations in domains CNB-A and CNB-B had minimal or no effect on PKG activation *in vitro* [19–22].

Despite these many insights, important aspects of the function of *Plasmodium* PKG remain unclear. Global phosphoproteome studies [23–27] have identified seven phosphorylation sites in *P. falciparum* PKG (PfPKG); two of these sites are found within CNB-B (T202, Y214) and five within the kinase domain (S576, Y694, T695, S817 and S819), including two residues within the activation loop of PKG (Y694 and T695). Activation loop phosphorylation is a mechanism exploited by several protein kinases to regulate activity, and indeed *in vitro* studies on mammalian PKG have shown that activation loop phosphorylation is a prerequisite for activity; in contrast, mutation of PfPKG T695 to Ala or Gln did not ablate kinase activity *in vitro* [19,28]. However, substitution of all seven phosphorylation sites with Ala rendered PfPKG inactive *in vivo* [13]. To date, it is unclear whether this effect is due to the absence of phosphorylation at these seven positions. The relative importance of these phosphorylated residues and how they regulate PfPKG function are unknown.

Here we have carried out a detailed examination of the role of the putative phosphosites in PfPKG function. Using a conditional allelic replacement approach, we show that saturation of CNB-B with cGMP is essential for full kinase activation and parasite egress. We further show that PfPKG activation loop phosphorylation occurs early in intraerythrocytic parasite development, prior to activation of the kinase at egress. Activation loop phosphorylation is not required for PfPKG enzyme activity but is a prerequisite for PfPKG function *in vivo* as it modulates the substrate specificity of the kinase. Our results highlight the unique manner in which PKG regulates the egress signalling cascade in the malaria parasite.

## Results

### Multiple phosphosite mutations destabilise the pentagonal architecture of PfPKG

We have previously shown that substitution of all seven of the known phosphosites of PfPKG (S1A Fig) with Ala residues is not tolerated by the parasite, producing a phenotype that mimics conditional disruption of the gene [13]. To examine these phosphosites in more detail, we used the previously established parasite line called *pfpkg_2lox* [13]. A marker-free Cas9-mediated mutagenesis strategy [29] was used to generate two parasite lines in which rapamycin (RAP)-induced, DiCre-mediated recombination between the two *loxN* sites would result in two distinct outcomes: either expression of an effectively wild-type form of PKG (line *pkgsynth_*GFP, S1B Fig) or a mutant PfPKG isoform in which all seven phosphosites are substituted with Ala residues (line *pkgΔPhos*, S2A Fig). In both cases, allelic replacement was expected to produce full-length PfPKG isoforms fused to the fluorescent protein eGFP [30].

Correct integration of the targeting constructs and excision of the floxed genomic sequences upon RAP-treatment were confirmed by PCR (S1C and S2A Figs). We then examined the two *P. falciparum* lines for their ability to replicate following excision of the floxed sequences. This confirmed that whereas RAP-treated *pkgsynth_GFP* parasites developed normally, RAP-treated ΔPhos parasites were not viable (S1D Fig), remaining trapped inside the host erythrocyte as previously described (S1E Fig and S1 and S2 Movies) [13]. Western blot analysis of RAP-treated ΔPhos schizonts revealed multiple PfPKG-derived species, pointing towards degradation of the protein. It was concluded that mutation of all seven phosphosites results in mis-folding and/or degradation of the protein (S1F Fig).

Y694 has a central role in the structural stabilisation of PfPKG [19]. Replacement of Y694 with Ala likely weakens the packing of the kinase activation loop against the long α-helix

(R149-D178) connecting CNB-A to CNB-B (S1G Fig). This is the reason we have excluded Y694 from subsequent analysis.

## cGMP saturation of CNB-B is essential for PKG activation *in vivo*

Using a similar conditional mutagenesis-based approach, we next examined the importance of phosphorylation of Y214, which is located close to the cGMP binding domain of CNB-B. We substituted Y214 with Ala by conditional allelic replacement (Fig 1A and S2B Fig), and expression of the mutant PfPKG was verified by Western blot analysis (Fig 1B). RAP-treated *pkgY214A* parasites did not proliferate (Fig 1C), while time-lapse video microscopy confirmed their inability to egress (S3 Movie). To better delineate whether this phenotype was due to deficient phosphorylation or structural defects due to the loss of the Tyr residue, we created a new line in which, upon conditional allelic exchange, Y214 was replaced by Phe (Fig 1A, 1D and S2B Fig), an amino acid with similar biophysical properties to Tyr but which cannot be phosphorylated. This mutation had no effect on parasite growth, suggesting that the previously observed growth arrest in Y214A parasites was not phosphorylation-dependent (Fig 1E). Since Y214 lies within CNB-B, albeit not within the cGMP-binding pocket, the lethal phenotype observed in Y214A parasites raised the question of whether inefficient cGMP binding could be preventing kinase activation. To address this, we monitored the appearance in parasite culture supernatants of SERA5, an abundant parasitophorous vacuole protein which at the point of egress is released in a proteolytically processed form into culture supernatants [10,31,32]. The assay was performed in the presence or absence of zaprinast, a phosphodiesterase inhibitor. Addition of zaprinast to schizonts induces egress by raising cGMP levels, leading to PKG activation and discharge of exonemes and micronemes [10]. As shown in Fig 1F (upper panel), SERA5 was absent from culture supernatants of the Y214A schizonts but appeared upon zaprinast treatment, showing that zaprinast overcomes the egress block imposed by the Y214A mutation. In contrast, zaprinast treatment of RAP-treated ΔPhos schizonts or schizonts lacking PfPKG expression entirely (line PKG_cKO) [15], did not result in release of processed SERA5 (Fig 1F, lower panel). Collectively, these results point towards a structural role for Y214 in the binding of cGMP to CNB-B and indicate that phosphorylation of this residue is not important for PfPKG function.

To address the question of whether a similar mechanism exists for binding of cGMP to CNB-A, we extended our studies to CNB-A by substitution of F96, which occupies the position equivalent to Y214 in CNB-B. Conditional mutagenesis of this residue to Ala had no effect on parasite viability (S3A and S3B Fig) suggesting that either full saturation of CNB-A is not essential for PfPKG activation or that destabilisation of this domain by the mutation is minimal. Collectively, these data highlight the requirement for full saturation of CNB-B for PfPKG activity *in vivo*.

## Recombinant PfPKG$_{Y214A}$ has decreased affinity for cGMP

To corroborate our *in vivo* findings, we next expressed PfPKG in both wild-type and Y214A recombinant forms in *E. coli* and measured the corresponding enzyme activities using a real-time fluorescence kinase assay [33] (S4A and S4B Fig). Phosphorylation of a substrate peptide (PKAtide) was measured by detecting changes in fluorescence upon the generation of ADP and its binding to a coumarin-labelled biosensor. While both forms of recombinant PfPKG displayed similar affinities for PKAtide (Figs 1G and S4C, S1 Data) and ATP (Figs 1H and S4C), we found the $K_A$ for cGMP (concentration at half maximal velocity) to be approximately 2-fold higher for PfPKG$_{Y214A}$ (~ 1.2 μM) compared to ~ 0.6 μM for the wild-type enzyme (Fig 1I). This result is fully consistent with the Y214A mutation subtly reducing the affinity of

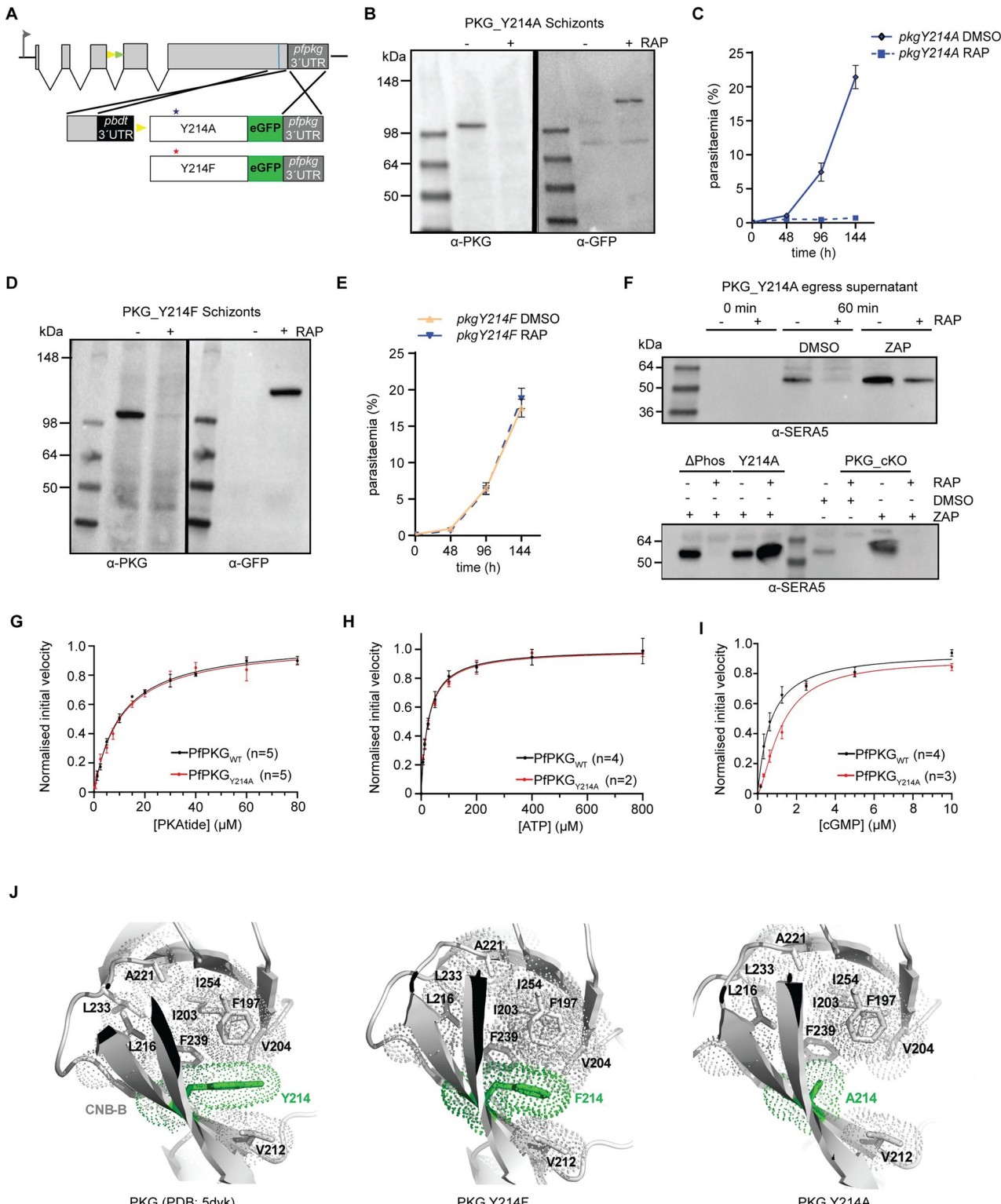

**Fig 1. Saturation of CNB-B with cGMP is a prerequisite for PKG activation and parasite development.** (A) Schematic of the Cas9-enhanced targeted homologous recombination approach used to create lines *pkgY214A*, *pkgY214F*. Blue line indicates the position targeted by the gRNA. Stars depict the relative positions of the mutated amino acids. **(B)** Western blot analysis of *pkgY214A* DMSO or RAP-treated schizonts showing expression of endogenous PfPKG or the mutant form fused to eGFP, respectively. **(C)** Replication rates over three cycles of line *pkgY214A*. Error bars, ± S.D (n = 3). **(D)** Western blot analysis of *pkgY214F* DMSO or RAP-treated schizonts. **(E)** Replication rates over three cycles of line *pkgY214F*. Error bars, ±

S.D (n = 3). **(F)** (Upper panel) Western blot of *pkgY214A* parasites, showing no release of SERA5 P50 into culture supernatants of RAP-treated schizonts consistent with impaired egress. Upon addition of zaprinast (ZAP), SERA5 P50 is detected in the supernatant suggestive of parasite egress. (Lower panel) Zaprinast treatment has no effect on ΔPhos RAP-treated schizonts nor PKG_cKO schizonts. **(G, H, I)** Determination of the PKAtide $K_m$, ATP $K_m$ and cGMP $K_A$ for recombinant PfPKG$_{WT}$ (in black) and PfPKG$_{Y214A}$ (in red). Means ± SEM (see S1 Data). **(J)** Cartoon representation of PfPKG CNB-B (PDB: 5DYK—cyan) with Y214, F214, A214 in green and the core hydrophobic residues in white. (left panel) Y214 participates in the overall hydrophobic core stabilization of the CNB-B domain. (middle panel) The PfPKG mutant Y214F still retains this stabilization role within the CNB-B core. (right panel) In the Y214A mutant, hydrophobic interactions are lost, likely loosening the compact fold of the CNB-B β-barrel.

the CNB-B domain for cGMP. Structural modelling suggests that the Ala substitution may disrupt stacking interactions performed by the Y214 phenyl ring (Fig 1J).

## Kinase domain phosphomutants disrupt PfPKG function *in vivo* but not *in vitro*

To assess the role in PfPKG function of the five remaining phosphosites (T202, S576, T695, S817, S819), four of which lie within kinase domain, we next generated conditional mutant line *pkg_STmut* (Figs 2A and S2B), in which all five sites were substituted with Ala. RAP-treatment of the *pkg_STmut* line to produce a quintuple mutant resulted in a growth arrest (Fig 2C). We performed egress assays in the presence or absence of zaprinast to examine whether the observed phenotype could be bypassed by increased cGMP levels. As shown in Fig 2D, processed SERA5 was completely absent from culture supernatants of RAP-treated *pkg_STmut* schizonts, indicating that in this case the arrested phenotype cannot be bypassed by elevated cGMP levels.

To gain a better understanding of the five phosphosites in kinase function, we recombinantly expressed the quintuple mutant (PfPKGSTmut—S4A and S4B Fig) to analyse its activity. Circular dichroism (CD) analysis showed that while recombinant PfPKG$_{STmut}$ possessed a slightly increased α-helical content relative to wild-type PfPKG (~ 24% versus 17% for wild-type PfPKG), the overall secondary structural composition of the proteins was similar, confirming the structural integrity of the mutant (Fig 2E and S1 Data). We next sought to compare the affinity of wild-type PfPKG and PfPKG$_{STmut}$ for the PKAtide substrate (Figs 2F and S4C). Both recombinant proteins were found to require similar concentrations of PKAtide (10 and 13 μM, respectively, S1 Data) to achieve half maximal velocity, suggesting that binding of the substrate is not significantly affected in PfPKG$_{STmut}$. Similarly, the $K_A$ for cGMP for PfPKG$_{STmut}$ was ~ 0.5 μM, within the range of that for wild-type PfPKG (Figs 2G and S4C), confirming that cGMP-dependent activation is also not affected. Attempts however to obtain reliable $K_m$ values for ATP with PfPKG$_{STmut}$, were unsuccessful, suggestive of subtle structural changes affecting ATP binding but not kinase activity *in vitro*.

Selective small molecule inhibitors of PKG (such as the imidazopyridines C2 and ML-10) are known to bind in a small hydrophobic pocket in close proximity to the ATP-binding site [19,34,35]. The inability to obtain $K_m$ values for ATP in the quintuple mutant prompted us to examine the effect of these two inhibitors on the kinase activity of PfPKG$_{STmut}$. As shown in Fig 2H, this mutant was less sensitive to both C2 and ML-10 compared to PfPKG$_{WT}$, confirming the presence of structural changes around or within the ATP binding pocket.

## N and C-lobe phosphomutants reduce sensitivity of PfPKG to ATP-competitive inhibitors

Collectively, the above results pointed towards a regulatory role for phosphorylation of PfPKG, raising the question of whether the observed phenotype is attributed to a single amino acid residue substitution or multiple changes. To address this, we created mutant parasite lines

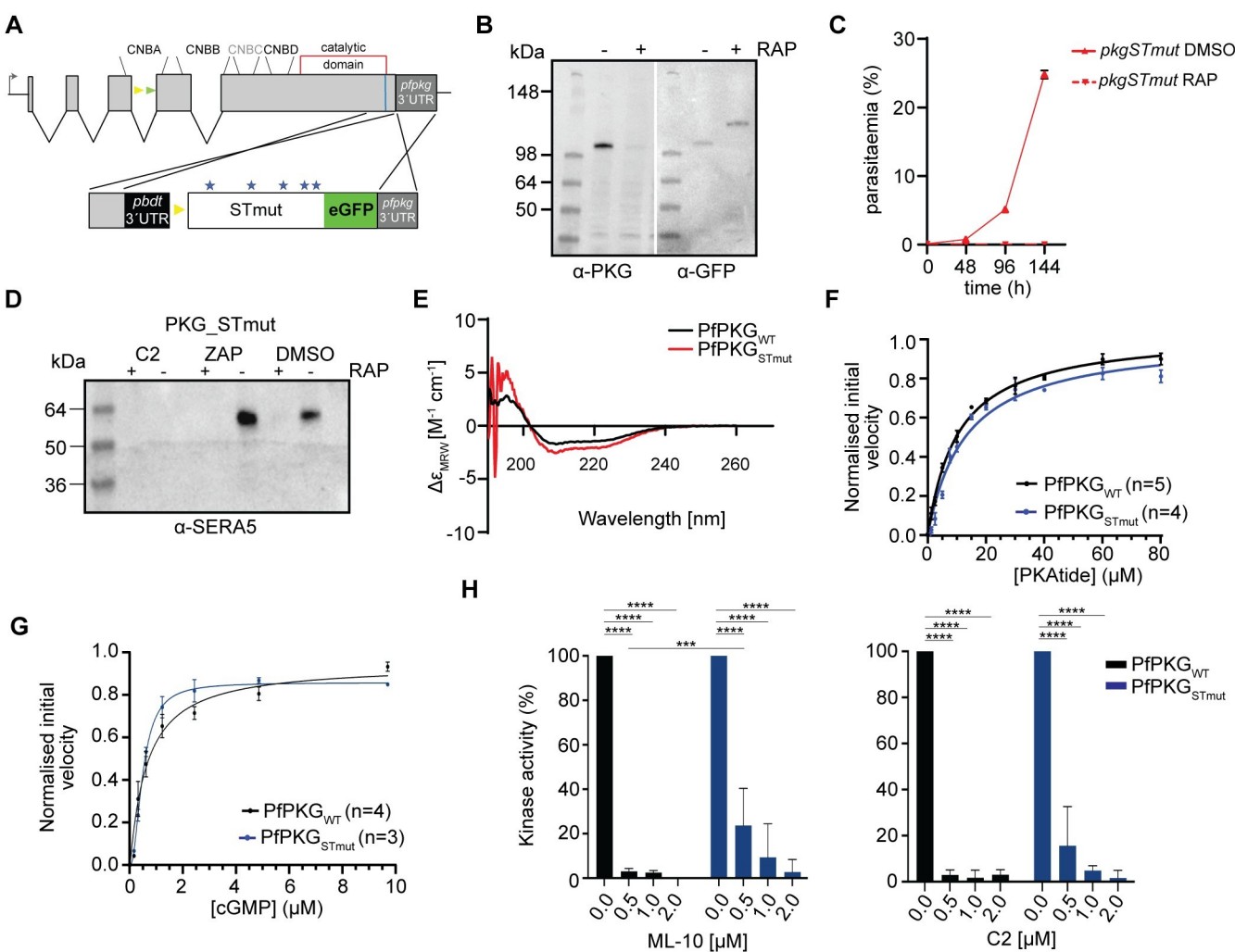

**Fig 2. Kinase domain phosphomutants render PfPKG inactive *in vivo* without affecting kinase activity *in vitro*.** (A) Schematic of the approach used to create line *pkgSTmut*. Blue line indicates the position targeted by the gRNA. Stars depict the relative positions of the mutated amino acids. (B) Western blot analysis showing the distinct PKG versions in the absence or presence of rapamycin (RAP). (C) Replication rates of line *pkgSTmut* over three cycles. Error bars, ± S.D (n = 3). (D) Western blot of *pkgSTmut* parasites, showing no release of SERA5 P50 into culture supernatants of RAP-treated schizonts even upon addition of zaprinast (ZAP) consistent with impaired egress. (E) Circular dichroism (CD) spectra of recombinant PfPKG$_{WT}$ and PfPKG$_{STmut}$, showing no significant changes in secondary structure. (F and G) PKAtide K$_m$ determination and cGMP K$_A$ determination for recombinant PfPKG$_{STmut}$ (in blue), relative to the control (PfPKG$_{WT}$ in black). Means ± SEM (see S1 Data). (H) Kinase activity of PfPKG$_{WT}$ and PfPKG$_{STmut}$ in the presence of the established PKG inhibitors ML-10 (left panel) and C2 (right panel). Means ± SD, Two-way ANOVA with P<0.0001 (****) and P = 0.0009 (***).

expressing single point mutations at each of the five phosphosites. In line *pkgT202A* (S3A Fig), correct integration and expression was confirmed by diagnostic PCR (S2B Fig) and Western blot, respectively (S3C Fig). The T202A mutation had no effect on parasite growth (S3D Fig), suggesting that phosphorylation of this residue, within CNB-B does not play a critical role in PfPKG function.

We next examined the effects on parasite development of single S576A (located in the kinase domain N-lobe), S817A and S819A (located in the C-lobe tether of the kinase domain C-lobe) mutants respectively, as well as the double mutant S817A_S819A (Fig 3A). For this, we took advantage of the distinct lox sites present in line *pkg_2lox*. DiCre-mediated recombination between the two *loxN* sites would lead to expression of a mutant eGFP-tagged PfPKG in which S576 is mutated to Ala, while recombination between the two *lox2272* sites would

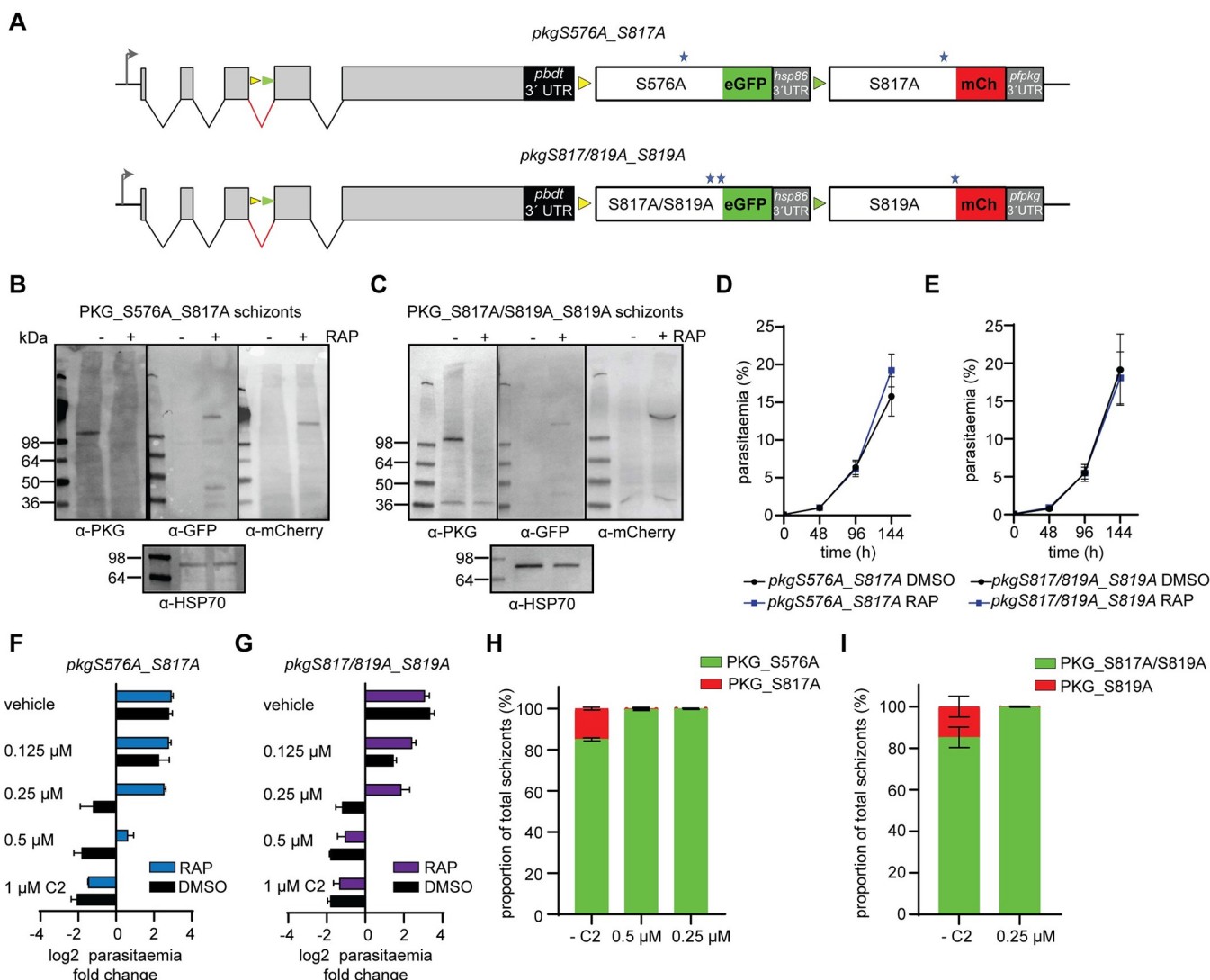

**Fig 3. Kinase domain mutations decrease the *in vivo* sensitivity of PfPKG to established ATP-competitive inhibitors. (A)** Schematic of lines *pkgS576A_S817A* and *pkgS817A/S819A_S819A*. The positions of the *loxN* (yellow arrowheads) and *lox2272* (green arrowheads) sites are indicated. **(B, C)** Western blot analysis showing loss of expression of PKG upon RAP-treatment of *pkgS576A_S817A* and *pkgS817A/S819A_S819A* schizonts, respectively, with appearance of signals corresponding to the eGFP fusions (PKG_S576A and PKG_S817A/S819A) and the mCherry fusions (PKG_S817A and PKG_S819A). **(D, E)** Replication rates over three cycles of lines *pkgS576A_S817A* and *pkgS817A/S819A_S819A*. Error bars, ± S.D (n = 3). **(F, G)** Parasitaemia fold change (log2) after 1 cycle, in the presence of different amounts of C2 or vehicle (DMSO). Error bars, ± S.D (n = 3). **(H, I)** Quantification by live imaging of the relative proportions of RAP-treated *pkgS576A_S817A* and *pkgS817A/S819A_S819A* schizonts (sampled at the end of cycle 2) in the presence of C2 (n = 2, more than 300 schizonts counted in each experiment).

instead lead to expression of an mCherry-tagged PfPKG where S817 is mutated to Ala. A similar approach was followed for mutants S817A_S819A and S819A (S5A and S2C Figs). Western blot analysis confirmed expression of both eGFP and mCherry tagged mutant forms of PfPKG in RAP-treated parasites (Fig 3B and 3C). Live microscopy of the eGFP and mCherry-positive parasites in both lines suggested that recombination between the *loxN* sites was favoured (S5B Fig), as the ratio between eGFP and mCherry expressing parasites was ~8:2, as previously seen [13]. Monitoring replication of DMSO- and RAP-treated parasites for both lines over three cycles revealed no significant difference in growth rates (Fig 3D and 3E). We also performed time-lapse video-microscopy of RAP-treated schizonts to examine the mutant phenotype at

the single-cell level. As shown in S5C Fig and S4 and S5 Movies, both PfPKG mutant lines underwent normal egress, demonstrating that these mutations have no effect on parasite development.

We next wanted to corroborate our *in vitro* findings, in which the quintuple mutant displayed reduced sensitivity to known ATP-competitive inhibitors of PKG and investigate whether we could mimic these results *in vivo*. For this, we performed growth assays of DMSO- and RAP-treated parasite lines *pkgS576A_S817A* and *pkgS817A/S819A_S817A* in the presence of various concentrations of C2. Both RAP-treated lines were able to grow in the presence of 0.25 μM C2, while the RAP-treated pkgS817A/S819A_S817A line was also able to grow in 0.5 μM C2. By contrast the DMSO-treated lines (i.e. wild type), could not grow at these concentrations of C2, as expected (Fig 3F and 3G). To examine which mutations are responsible for parasite proliferation in the presence of C2, we allowed RAP-treated parasites to develop for two complete intraerythrocytic cycles and then examined them microscopically for eGFP or mCherry expression. Our results showed that mutant S576A can grow in the presence of 0.5 μM and 0.25 μM C2 while the double mutant S817A/S819A can proliferate in the presence of 0.25 μM C2 (Fig 3H and 3I). These results suggest that while these kinase domain mutations have no effect on parasite viability, they do alter the sensitivity of PfPKG to established ATP-competitive inhibitors.

### Activation loop phosphorylation of PfPKG is essential *in vivo* but not *in vitro*

In light of the above results, we next focused our studies on T695, which resides in the activation loop of PfPKG [19]. In many protein kinases of the AGC family, phosphorylation of a Thr residue in the activation loop converts the enzyme to its active state. In the published x-ray crystal structure of recombinant PfPKG, T695 is unphosphorylated and the activation loop adopts an active conformation in the absence of cGMP. In that study, phosphorylation of T695 was found to occur only in the presence of ATP, $Mg^{2+}$ and cGMP, while Ala or Gln substitutions of T695 did not affect the activity profile of the kinase *in vitro* [19].

To examine the *in vivo* role of phosphorylation of T695, we created both a conditional phosphomutant (T695A) and a putative phosphomimetic (T695E) *P. falciparum* line (Figs 4A, 4B and S2B). Remarkably, neither mutation was tolerated by the parasite (Fig 4C and 4D) resulting in both cases in a phenotype characteristic of loss of PKG function, as indicated by development to mature schizonts that failed to undergo egress (S6 and S7 Movies). This result prompted us to examine the phenotype in more detail to investigate whether it mimics conditional disruption of PfPKG [13,15]. The egress arrest was not overcome by the presence of zaprinast (Fig 4E). We then examined release and translocation to the merozoite surface of the micronemal protein AMA1, which occurs at or just prior to egress and is linked to PfPKG activation [10]. Comparative IFA of DMSO- and RAP-treated schizonts, showed a severe defect in AMA1 translocation in both the PKG_T695A and PKG_T695E mutants (Fig 4F), consistent with them being defective in microneme discharge.

We have previously shown that *Plasmodium* PKG interacts with a transmembrane protein (called ICM1) [15]. To examine whether the mutations affect that interaction, we used antibodies against GFP to perform pull down analysis of RAP-treated schizonts of lines *pkgT695A* and *pkgT695E* (both GFP-fusions) and WT parasites (B11) alongside a previously described line in which wild-type PKG is fused to eGFP [15]. Mass spectrometric examination of the pull-downs showed that both PKG_T695A and PKG_T695E interact strongly with PfICM1 (Fig 4G and S2 Data). This suggests that the egress defect in both mutants is not a result of disruption of the PKG-ICM1 interaction.

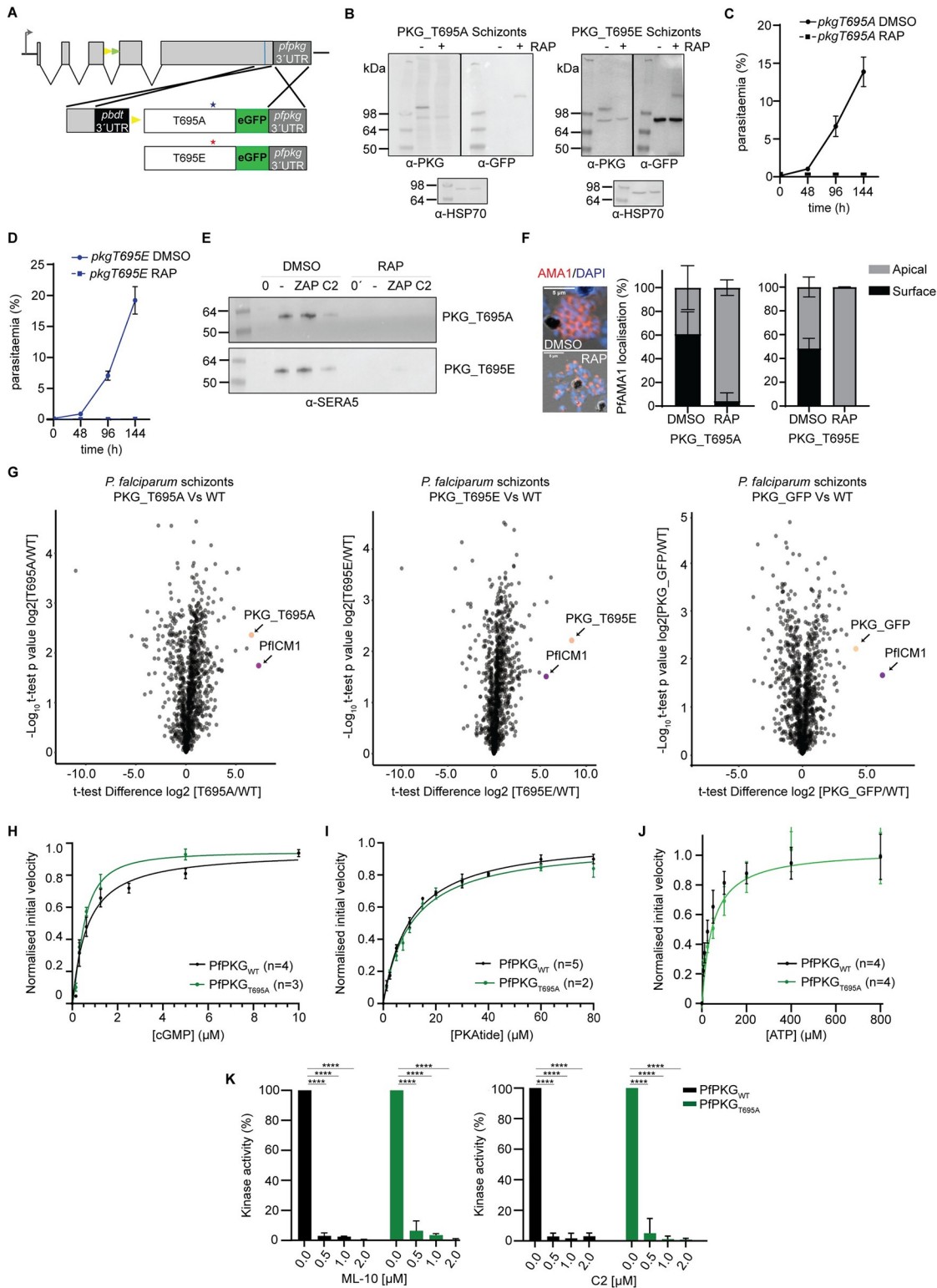

**Fig 4. Phosphomutant and phosphomimetic modifications of T695 interfere with PfPKG function in the parasite but do not affect kinase activity.** (**A**) Schematic of the approach used to create lines *pkgT695A* and *pkgT695E*. Blue line indicates the position targeted by the gRNA. Stars depict the relative positions of the mutated amino acids. (**B**) Western blot analysis of PKG_T695A and T695E schizonts in the absence or presence of rapamycin (RAP). HSP70 was used as a loading control (**C, D**) Replication rates over three cycles of lines *pkgT695A* and *pkgT695E*. Error bars, ± S.D (n = 3). (**E**) Egress assay of parasites *pkgT695A* and *pkgT695E*,

showing no release of SERA5 P50 into culture supernatants of RAP-treated schizonts even upon addition of zaprinast (ZAP) consistent with impaired egress. **(F)** Representative IFA image of *pkgT695A* parasites showing translocated AMA1 in DMSO-treated merozoites or micronemal AMA1 in RAP-treated merozoites. Right: Quantification of AMA1 relocalization in both *pkgT695A* and *pkgT695E* (n = 2, >150 schizonts were quantified in each). Values, means ± SD. **(G)** Volcano plots showing quantification of proteins identified by quantitative MS in pull-downs from PKG-GFP, PKG_T695A and PKG_T695E compared to WT (control) parasites (n = 2). Significance (Student's t test) is expressed as log10 of the P value (y axis). Enrichment of interaction partners compared to controls (x axis). PKG is indicated (magenta dot) as well as ICM1 (orange dot). **(H, I and J)** cGMP $K_A$ determination, PKAtide $K_m$ determination and ATP $K_m$ determination of recombinant PfPKG$_{T695A}$ (in green), relative to the control (PfPKG$_{WT}$ in black). Means ± SEM (see S1 Data). **(K)** Kinase activity of PfPKG$_{WT}$ and PfPKG$_{T695A}$ in the presence of the established PKG inhibitors ML-10 (left panel) and C2 (right panel). Means ± SD, Two-way ANOVA with P<0.0001 (****).

To validate the previously published observations on PKG_T695A kinase activity (as the enzyme data were not presented in that study) [19], we recombinantly expressed the T695A mutant in *E.coli* (PfPKG$_{T695A}$). Kinetic assays confirmed the kinase activity of this mutant, with no differences in the $K_A$ for cGMP and $K_M$ for substrate binding relative to PfPKG$_{WT}$. The only difference we observed was the $K_M$ for ATP in PfPKG$_{T695A}$ was 44.6 μM compared to 25.9 μM for PfPKG$_{WT}$ (Figs 4H, 4I, 4J and S4A–S4C), S1 Data, We also examined the sensitivity of this mutant to ATP-competitive inhibitors and observed no differences relative to PfPKG$_{WT}$ (Fig 4K).

Active site loop phosphorylation coordinates defined kinase regions (C-helix, catalytic loop and activation loop) via a hydrogen bonding network [36,37]. To examine if the activation loop of PfPKG shares structural similarities with well characterised members of the same family, the crystal structures of PfPKG (PDB: 5dyk) [19] and mouse PKA (PDB:1atp) [38] were overlayed (rmsd: 1.3Å) to look at the amino acid conservation within the activation loop phosphorylation site. As can be seen in S6 Fig, two out of the five residues involved in the stabilization of the phosphate group via hydrogen bonding in PKA are not conserved in PfPKG. Residues H87 from the C-helix and T195 from the activation loop in PKA are replaced in PKG by N585 and A693 respectively.

Together these results suggest that structural modifications of T695 that either prevent phosphorylation at this site or that mimic constitutive phosphorylation, have no impact on kinase activity but fatally interfere with PfPKG function in the parasite.

## The egress defect in PfPKG_T695A and T695E parasites is associated with aberrant phosphorylation

Early studies on mammalian PKGIα showed that substitutions of the activation loop Thr (T517) are detrimental to kinase activity *in vitro* [28]. Our *in vivo* results suggested that phosphorylation of the PfPKG activation loop might function in a novel way to regulate essential processes in the malaria parasites. To examine the temporal profile of PfPKG T695 phosphorylation across the intraerythrocytic life cycle we exploited a parasite line expressing PfPKG fused to a C-terminal triple hemagglutinin tag (HA) [12]. PfPKG-HA was affinity-purified from extracts of highly synchronised trophozoites (34 h post invasion) and schizonts (46 h post invasion) (S7A Fig). Mass spectrometric analysis identified the phosphopeptide corresponding to T695 (AYpTLVGTPHYMAPEVILGK) in tryptic digests of both extracts (S7B Fig and S3 Data). These data suggest that T695 is phosphorylated well before the point of PKG activation at egress.

To further explore the functional consequences *in vivo* of the T695A and T695E mutations, we determined their impact on the global schizont phosphoproteome. For this, we compared RAP-treated schizonts of lines *pkgT695A* and *pkgT695E* (both GFP-fusions) to schizonts from a line expressing wild-type PfPKG tagged with eGFP [15]. Both mutations produced a profound effect on the schizont phosphoproteome with more than 400 phosphosites being

differentially phosphorylated relative to the wild-type parasites (Fig 5A and S4 Data). Using the same criteria for significant changes in phosphorylation, we compared the phosphorylation profile obtained from the T695 mutants with the previously published PKG null phosphoproteome (Fig 5B) [15]. Although there were more than 1000 hypophosphorylated sites detected in the PKG knockout, compared to only 234 and 189 respectively in the T695A and T695E mutants, the number of detected hyperphosphorylated sites was more than doubled in both the T695A and T695E mutants compared to the knockout. Focusing on PfPKG phosphorylation sites, we identified only three sites in this analysis, which were all differentially phosphorylated. T695 was, as expected, significantly hypophosphorylated in both mutants as was T699, a second Thr in the active site loop. In contrast, S819 was hyperphosphorylated in both mutants (Fig 5C). Interestingly, in the T695A phosphoproteome we identified a single hyperphosphorylated site in ICM1 (S4 Data). Collectively the observed deregulated phosphosites included motifs in many proteins known to be involved in invasion and egress including the micronemal protein AMA1, the egress effector protease SUB1, the merozoite surface protein MSP1, PKAc, MyoE and CDPK5 (S4 Data). Motif analysis identified two significantly enriched motifs; a motif in T695A hyperphosphorylated phosphopeptides with strict preference for Arg in position -3 and a motif in T695E hyperphosphorylated phosphopeptides with a strict preference for Lys in positions -3 and -2 (Fig 5D); both resemble minimal PKA/PKC motifs.

To gain a better understanding of these biological signatures, we mapped the preferentially phosphorylated proteins of each mutant to protein–protein association networks in the STRING database (S8 Fig). MCL clustering identified two clearly enriched protein clusters of ATPase activity and mRNA binding/splicing (Fig 5E). Gene ontology (GO) term enrichment analysis on hyperphosphorylated proteins of both mutants suggested deregulation in phosphatidylcholine biosynthetic processes and mRNA binding, similar to the PKG null phosphoproteome (S9 Fig). Surprisingly, hypophosphorylated proteins of mutant T695E showed only basal deregulation of Molecular Function GO enrichment, while in the T695A mutant, ATPase−coupled transmembrane transporter activity was affected (S9 Fig).

## T695A and T695E mutations alter the substrate preference of PfPKG *in vitro*

In kinases of the AGC family, the activation loop is part of the peptide substrate binding surface. The differentially phosphorylated peptides observed in both T695A and T695E phosphoproteomes raised the question of whether these mutations could alter the substrate specificity of PfPKG. To examine this we used an oriented peptide array (OPAL) library in which one of the 20 naturally occurring amino acids was fixed at each of the eight positions surrounding the phosphorylated residue (S or T). The remaining positions were made up of approximately equimolar amounts of the 16 amino acids (excluding Cys, Ser, Thr and Try) [39].

We recombinantly expressed in *E. coli* the T695E mutant (PfPKG$_{T695E}$) to be included in our analysis alongside recombinant PfPKG$_{T695A}$ and PfPKG$_{WT}$ (S4A Fig). After confirming kinase activity for PfPKG$_{T695E}$ using ADP-Glo (S4D Fig) we screened the OPAL libraries with the three kinases (Fig 6). All three showed a preference for Thr compared to Ser as the phosphoacceptor, in agreement with previously published data [40]. In the case of PfPKG$_{WT}$ we observed a strong preference for basic residues in all positions flanking the phosphorylated residue, while acidic residues or Pro were not favoured throughout, particularly in the +1 position (Figs 6 and S10). In the case of both PfPKG$_{T695A}$ and PfPKG$_{T695E}$, when Thr was the phosphorylated residue no major differences from the wild type kinase were observed regarding the overall substrate preference and the bias against acidic residues in all positions. His was slightly preferred in positions -4 to -1 relative to the wild-type but Arg and Lys remained the

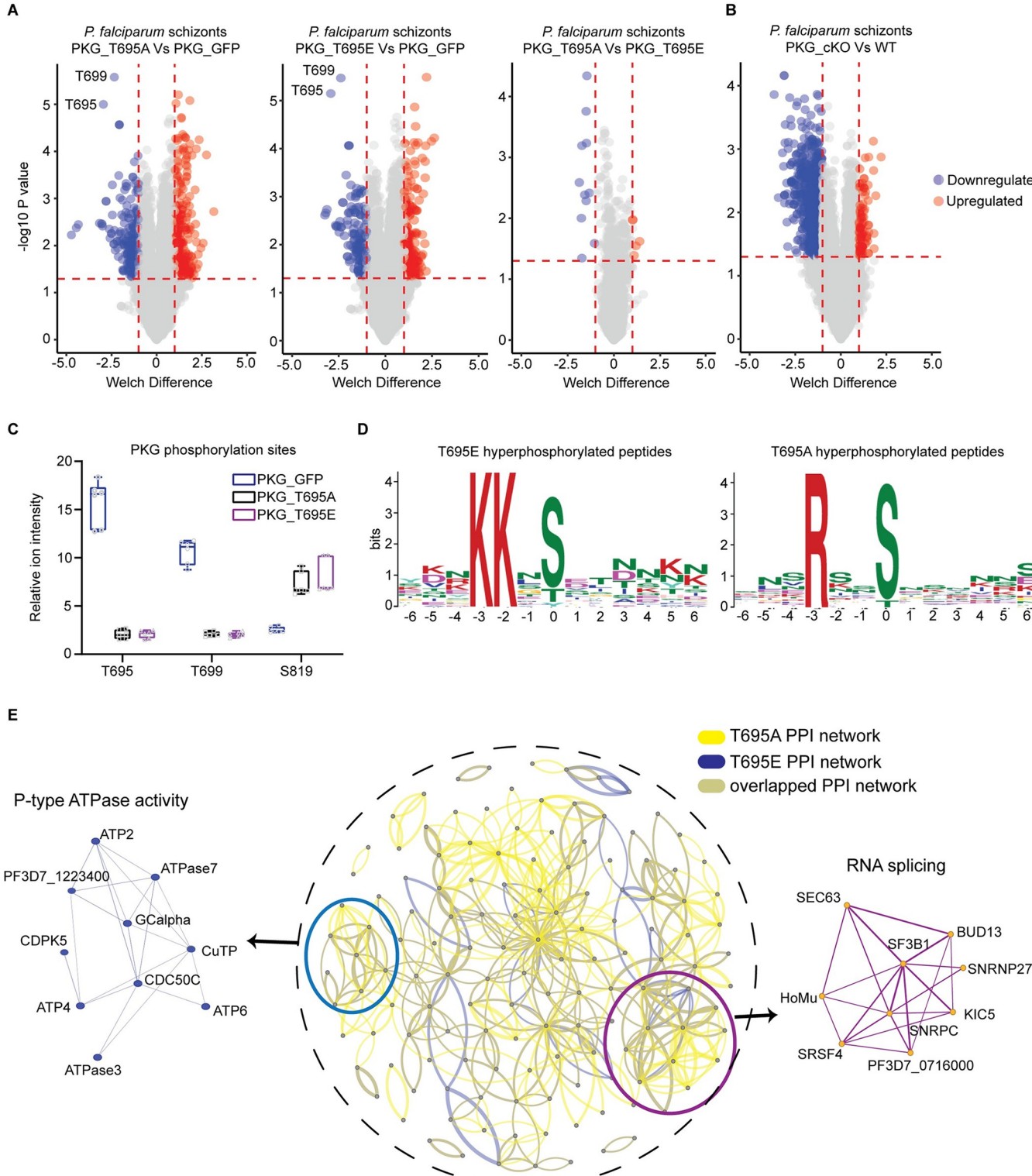

**Fig 5. Impact of T695A and T695E mutations on the schizont phosphoproteome. (A)** Volcano plots showing distribution of phosphopeptide abundance in T695A and T695E mutant parasites relative to PKG-GFP (WT) and between T695A and T695E. Blue circles correspond to significantly hypophosphorylated peptides and red circles to significantly hyperphosphorylated peptides. A cutoff of Welch difference of $<-1$ and $>1$ with P $<0.05$ and a localization probability of $>0.7$ represent significantly regulated peptides. **(B)** Volcano plots showing distribution of phosphopeptide abundance in PKG_cKO parasites relative to WT (data adapted from Balestra et al, 2021 (*15*)). **(C)** Box plots showing relative ion intensity of the 3 PfPKG phosphosites identified. The box plots depict the mean

(horizontal bar) and variance (min and max values). **(D)** Motif analysis of the significantly deregulated phosphosites in T695A and T695E parasites. Position 0 corresponds to the phosphorylated residue. **(E)** Protein interaction network of differentially regulated proteins in T695A (yellow lines) and T695E (blue lines) parasites. Inset circle shows the complete protein interaction network of the proteins (nodes) that had interaction evidence in STRING database (edges) (overlapped interactions shown in green). Thickness of the edge denotes confidence of prediction. Two clusters could be determined: RNA splicing (magenda) and P-type ATPase activity (light blue).

key amino acids (Fig 6). Interestingly, when Ser was the phosphorylated residue, we observed for both mutants a more relaxed specificity for Thr in positions -1 to +4, which was equally preferred to Arg and Lys (Fig 6).

Together, the OPAL library screens indicated subtle differences in substrate preference for both the T695A and T695E mutants relative to the WT kinase suggesting a role for phosphorylation of the activation loop T695 residue in substrate recognition.

## Discussion

PKG is the only cGMP effector in *Plasmodium* parasites, having an essential role throughout the life cycle and has consequently been heavily interrogated as a multi-stage drug target [34,41,42]. However, there are still numerous unanswered questions regarding the biology, regulation and physiological substrates of this essential enzyme. To gain an improved understanding of its biology in the asexual blood stages, we here examined the role of the seven previously identified PfPKG phosphosites on PfPKG activation and regulation.

We have previously shown that simultaneous mutation of all seven sites to Ala (our ΔPhos mutant) is detrimental to parasite survival [13]. In the present work we obtained clear evidence that this set of mutations leads to destabilisation of the protein. This is consistent with the recent finding by others that mutation of Y694 to Ala has a profound effect on protein expression in recombinant form and the mutant kinase is inactive [19]. The aliphatic chain of K157, which lies within the α-helix connecting CNB-A to CNB-B, performs a stacking interaction with the aromatic ring of Y694. The hydroxyl group of Y694 is engaged in hydrogen bonding with S160 Oγ. Substitution of Y694 with an alanine would remove these key stabilizing interactions and possibly release the PKG activation loop from the pentagonal frame, consequently affecting PKG function. The PKG destabilisation we observed in ΔPhos parasites is likely attributed to the Y694 mutation as no other mutant in our study was similarly destabilised. The lethal phenotype observed in ΔPhos parasites might therefore not be solely phosphorylation-dependent but can be attributed, at least in part, to a key structural role for Y694.

PfPKG possesses two potential phosphorylation sites in CNB-B. While no essential role was assigned to T202, mutation of Y214 to Ala provided us with a very interesting phenotype. Y214A parasites failed to egress but this blockade was overcome by treatment with a phosphodiesterase inhibitor (zaprinast), which is known to result in elevated intracellular cGMP levels. Previous *in vitro* studies have suggested that CNB-D is the critical domain for PfPKG activation with the highest affinity (40 nm compared to 1 μM for CNB-A and B) and selectivity for cGMP, while mutations in other CNB domains had minimal to no effect on kinase activation [20–22]. The exact mechanism by which Y214A parasites overcome the lethal phenotype, in the presence of zaprinast, is not clear. We hypothesize that either the increase in cGMP levels is sufficient to initiate the necessary structural changes in CNB-B or that CNB-B activation might not be essential, but the increase enables binding of cGMP to CNB-A and release of the kinase domain. Irrespective of the molecular mechanism, our findings highlight the need for co-operation between the different CNB domains and support the structural-relay model of cooperative activation proposed by El Bakkouri et al [19]. Y214 lies within a β-strand (V212-E218) of the CNB-B β-barrel, with its hydroxyl group facing outwards and being

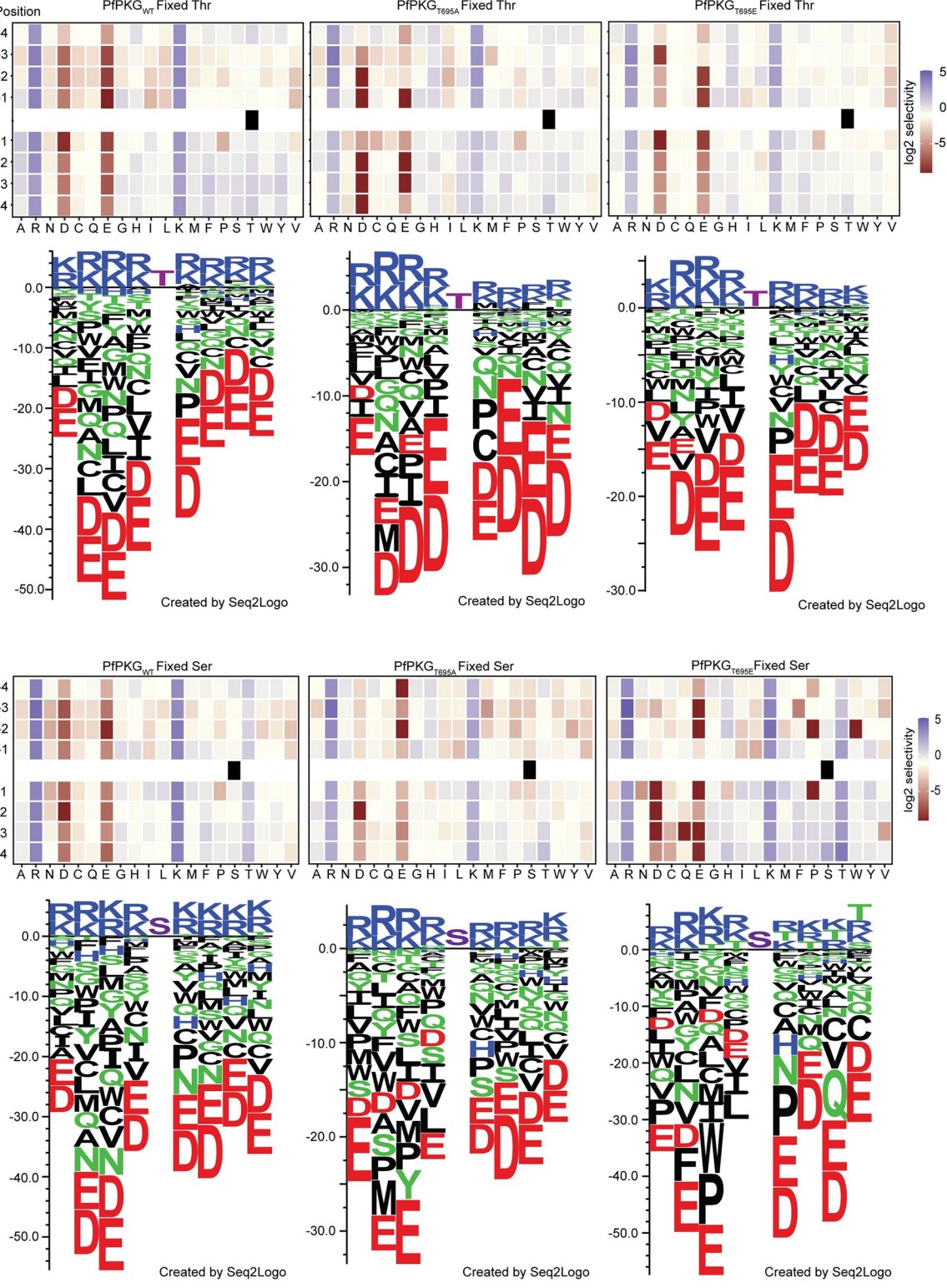

**Fig 6. T695 mutations alter the substrate specificity of PfPKG.** Heatmaps showing the specificities of PfPKG_WT, PfPKG_T695A and PfPKG_T695E, measured using the X4-S/T-X4 OPAL library (n = 1). Upper and lower panels are Fixed Threonine and Fixed Serine libraries, respectively. Enrichment scores were log2-transformed and displayed as follows: blue (favoured), beige (neutral), red (disfavoured). Sequence logos obtained are shown below each heatmap. Values were converted into a position-specific scoring matrix (PSSM) and Seq2Logo 2.0 was used to convert this to amino acid preferences (shown as sequence bits). Average level was set to 0 and positive and negative values show amino acids favoured or disfavoured in that position. Fixed Ser and Thr displayed in magenta colour.

accessible to the solvent while the phenyl ring interacts with residues I203, V204, V212 and F239 to stabilise the hydrophobic core of the CNB-B domain. Substitution of Y214 with Ala is predicted to disrupt the intramolecular stacking interactions present in CNB-B (Fig 1J). The phenyl ring of Y214 is mainly stabilised by pi-stacking interactions with F239 (parallel displaced) and by hydrophobic interactions involving the side chains of I203, V204 and V212. These core stabilizing interactions could still be maintained in the Y214F mutant due to the presence of its aromatic ring. By contrast, in the compact Y214A aliphatic mutant these interactions would be lost, leading to an intrinsic structural destabilisation of the CNB-B domain (Fig 1J). As a result, the compactness of the CNB-B β-barrel would be affected and the fold locally relaxed, which in turn could modify the affinity of cGMP for the CNB-B domain. Our *in vitro* studies supported this notion as the $K_A$ for cGMP was 2-fold higher in *Pf*PKG$_{Y214A}$ than in PfPKG$_{WT}$. While not significant *in vitro*, this subtle change in cGMP affinity appears to be important in Y214A mutant parasites.

Our phosphomutant analysis has determined that substitution of S576 and S817/S819 can affect binding of ATP-competitive inhibitors. Targeting PfPKG with small compounds has been a subject of significant research efforts, so potential resistance of the kinase to inhibitors is an important issue. Interestingly neither residue is in close proximity to the binding pocket of inhibitor ML10 (and probably C2) as was shown by the co-crystal structure of PvPKG with ML10 [34]. However, the available crystal structures and subsequent docking studies of the kinase domain are all based on the apo-form of the enzyme. So far, all attempts to obtain a complete PKG structure (in any organism) with cGMP bound have failed and therefore it is not possible to model the structural alterations of the N- and C-lobes following cGMP binding to examine how S576 and S817/S819 influence ATP binding. It is however worth emphasizing that previous attempts to generate *P. falciparum* parasites resistant to PKG inhibitors have failed [34, 43] suggesting that mutations affecting the ATP binding pocket are not well tolerated by the parasite.

Our phosphoproteome analysis identified a previously undescribed phosphorylation site in the activation loop (T699). It would be of interest to examine whether mutation of this residue is lethal *in vivo* and whether it alters the substrate specificity of PfPKG. In both our PfPKG T695A and T695E mutants we noted an increase in phosphorylation of S819. Future studies will address whether this was a direct consequence of the altered substrate specificity of both mutants or an indirect downstream effect mediated by another kinase or phosphatase.

A surprising aspect of our study was the implied role of activation loop phosphorylation in substrate binding. A previous study suggested that in the apo enzyme, T695 is not phosphorylated and mutations of this site do not interfere with kinase activity [19] but here we showed that *in vivo* T695 is phosphorylated prior to kinase activation. Ala or Glu substitution of T695 *in vitro* had no significant impact on kinase activation but the same mutations were lethal *in vivo* as they seem to interfere with substrate specificity. This is a unique feature amongst mammalian PKGs and amongst the related cAMP-dependent protein kinases (PKAs). In both cases, lack of phosphorylation in the active site loop (by phosphorylation of T517 and T197, respectively) profoundly reduces kinase activity [28,36,44,45].

Our finding that PfPKG activation loop phosphorylation is not essential *in vitro* but has significant effects *in vivo*, closely mimics properties of PKCδ. PKC belongs to the AGC family and there are 12 isoforms with important functions in eukaryotic cells, although PKC is absent in *Plasmodium* parasites [46]. Most PKC isoforms are activated upon phosphorylation of the activation loop. However in PKCδ lack of active site loop phosphorylation (T507) does not affect kinase activity *in vitro* but has functional consequences in cells [47,48]; the T507A mutant of PKCδ can induce apoptosis similar to the WT kinase but cannot induce AP-1 and NF-κB reporter activation [47]. Although the substrate specificity is not altered between WT

PKCδ and the T507A mutant, the mutant kinase was inhibited in higher concentrations of specific substrates. In the case of PfPKG, we show here that mutations of the active site residue T695 affect substrate preference. As we have shown (S6 Fig), the PKA residue network that stabilizes the phosphate group in the phosphorylation site is not fully conserved in PKG. Two out of the five residues involved in the stabilization of the phosphate group via hydrogen bonding in PKA are not conserved in PKG, since H87 and T195 in PKA, are replaced by N585 and A693 respectively. Interestingly these two residues (H87 and T195) are also not conserved in PKCδ and replaced by T394 and A505 respectively. These amino acid substitutions, particularly T195 (PKA) to A693 (PKG) or A505 (PKCδ) might contribute to a different activation loop involvement in activation of these two kinases.

Unfortunately to date no direct substrates of PKG have been identified *in vivo*. Also, the consensus substrate phosphorylation motif of PfPKG is very similar to PfPKA making the distinction of physiological substrates very difficult. We hypothesize that *Plasmodium* PKG has many direct substrates, which we have yet been unable to identify. The global phosphoproteome studies highlight the importance of *Plasmodium* PKG as a signalling hub involved directly or indirectly in the regulation of diverse pathways [15,43,49,50]. We were not able to correlate our phosphoproteome data with the OPAL substrate specificity data for enrichment of phosphopeptides with Threonine in positions +1 to +4. This could be attributed to phosphopeptides directly phosphorylated by PKG not been enriched significantly. However our dataset paves the way for future experiments with libraries of differentially phosphorylated peptides identified from our PKG mutants. These experiments could potentially provide more insights into the substrate preference of PfPKG and whether activation loop mutants are subject to product inhibition, as is the case for PKCδ, possibly enabling the identification of physiological substrates. Future studies should explore whether activation loop phosphorylation is an autophosphorylation event or whether it is mediated by the *P. falciparum* orthologue of 3-phosphoinositide-dependent protein kinase-1 (PfPDK1 –Pf3D7_1121900), which has been shown to phosphorylate the PfPKAc activation loop (T189) [51] as is the case with many AGC kinases [52]. A structure of PfPKG with cGMP bound will be invaluable for understanding in detail the interactions of the different domains in the activated enzyme and how phosphorylation regulates this essential enzyme.

In conclusion, we have produced insights into the unique manner in which PfPKG is regulated in asexual blood stage parasites. Our in-depth analysis has identified a single phosphosite within the activation loop that is essential for the regulation of this important kinase. Surprisingly, we find that activation loop phosphorylation is not a mechanism for kinase activation *per se* but a way to potentially identify and prioritise phosphorylation of its physiological substrates to transduce signals necessary for the parasite to complete its life cycle.

## Materials and methods

### Reagents and antibodies

Rapamycin (Sigma) was used at 20 nM to treat parasites. The PKG inhibitor (4-[7-[(dimethylamino)methyl]-2-(4-fluorphenyl)imidazo[1,2-α]pyridine-3-yl]pyrimidin-2-amine (Compound 2) was used at 1 µM in cultures. Zaprinast (Sigma) was used at 75 µM in egress assays. Polyclonal human PKG antibodies (Enzo Life Sciences) were used for PfPKG detection (1:1000 or 1:2000, respectively for recombinant PfPKG). Anti-GFP mAbs (Roche) were used at a dilution of 1:1000, as was a polyclonal anti-mCherry (ab167453; Abcam). Polyclonal anti-SERA5 was used at 1:2000 [53] and anti-HSP70 (1:2000) was a kind gift of Dr Ellen Knuepfer. Anti-His mAbs (ThermoScientific) were used at 1:10:000 dilution to detect recombinant PfPKG

proteins. Restriction enzymes were from New England BioLabs (NEB) and DNA ligations were performed using the Rapid DNA Ligation kit (Roche).

### *Plasmodium falciparum* culture, transfection, synchronisation

Line *pfpkg_2lox* was maintained at 37°C in human RBCs in RPMI 1640 medium containing Albumax II (Thermo-Scientific) supplemented with 2 mM L-glutamine and was used for all genetic modifications described. Cultures were microscopically examined on Giemsa-stained thin blood films and mature schizonts were isolated by centrifugation over 70% (v/v) isotonic Percoll (GE Healthcare Life Sciences) cushions. Transfections were performed as previously described [54]. In brief, ~$10^8$ Percoll-enriched schizonts were resuspended in 100 µL of P3 primary cell solution containing 20 µg of CrispR/Cas9 plasmid and 60 µg of linearised donor plasmid. Program FP158 of the Amaxa 4D Nucleofector X (Lonza) was used for electroporation. 24h post transfection, the growth medium was supplemented with 2.5 nM WR99210 (Jacobus Pharmaceuticals New Jersey, USA).

All transgenic parasite clones were obtained by limiting dilution cloning in microplates. Parasite genomic DNA (gDNA) for genotype analysis was extracted using the Qiagen DNeasy Blood and Tissue kit and PCR analysis was done with CloneAmp HiFi PCR Premix (Takara) or Phusion polymerase (NEB).

DiCre activity was induced as previously described [54]. Samples for excision PCRs were collected 24 h post RAP treatment and for Western blot analysis at 42 h post RAP treatment.

### Plasmid construction and genotyping of transgenic lines

To create line *pkgsynth_GFP*, vector pHHI-PfPKG-GFP [15] was digested with SacI/NotI. The fragment comprising of the synthetic *pfpkg* fused to eGFP [30] and the *pfpkg* 3´UTR was cloned to vector pCR-Blunt (ThermoFischer Scientific) previously digested with the same enzymes. This vector was subsequently digested with SacI/AgeI to introduce the following sequence from plasmid pDC_loxnPKG:lox22mCherry [13]: 5´ homology arm, the PbDT 3´ UTR, a fragment comprising *loxN*, the 3´ 46 bp of the *sera2* intron and a synthetic fragment of *pfpkg* starting from exon 4. Final plasmid (pZB-PKGsynth-GFP) was linearised overnight with HindIII and transfected together with the pDC2-pkg Cas9 plasmid [13] into *P. falciparum* line *pfpkg_2lox*. Correct 5´ and 3´ integration was verified by PCR using primers wtpkg_For/ 5int_Rev and PKGsynth_For/3int_Rev respectively (List of primers used in this study are shown in S1 Table). Modified endogenous locus was confirmed using primers exon1_For/ PKGutr_Rev.

To create line *pkgΔPhos*, the *pfpkg* fragment from vector pZB-PKGsynth-GFP was replaced with the mutant form previously described [13] resulting in vector pZB-PKGΔPhos-GFP. This was linearised overnight with HindIII and transfected together with the pDC2-pkg Cas9 plasmid into *P. falciparum* line *pfpkg_2lox*. Correct integration was confirmed as described above and presence of mutations was confirmed by Sanger sequencing.

Lines *pkgT202A*, *pkgY214A*, *pkgY214F* and *pkgF96A* were created by overlapping PCR mutagenesis.

**pkgT202A.** This line was created using as template plasmid pZB-PKGsynth-GFP and primers St1_For/T202A_Rev, T202A_For/St1_Rev. 0.2 µL of each PCR product were mixed and used as template for amplification of the final fragment with primers St1_For/St1_Rev. Final PCR product was digested with HpaI/BsmI and ligated to plasmid pZB-PKGsynth-GFP previously digested with the same restriction enzymes. Mutation in resulting plasmid pZB-T202A-GFP was confirmed by Sanger sequencing.

***pkgY214A*, *pkgY214F* and *pkgF96A*.**   Same approach as described above was implemented for these plasmids, using primers St1_For/Y214A_Rev—Y214A_For/St1_Rev, St1_For/Y214F_Rev—Y214F_For/St1_Rev and St1_For/F96A_Rev—F96A_For/St1_Rev respectively. Final PCR product was amplified using primers St1_For/St1_Rev.

For line *pkgSTmut*, vector pZB-PKGΔPhos-GFP was used as a template for modifications. Primer sets St4_For/A694Y_Rev, St4_Rev/A694Y_For and St4_For/St4_Rev were used to mutate A694 to the original Tyrosine. Subsequently A214 was mutated back to Tyrosine using primers St1_For/A214Y_Rev and A214Y_For/St1_Rev. Final pcr fragment was amplified using primers St1_For/St1_Rev. Plasmids pZB-T695A-GFP and pZB-STmut-GFP were linearised overnight with HindIII and transfected independently alongside plasmid pDC2-pkg in *P. falciparum* line *PFpkg_2lox*.

Lines *pkgT695A* and *pkgT695E* were created by incorporating a fragment of synthetic *pfpkg* harbouring the relevant mutation (commercially obtained as a gBlock from IDT) into plasmid pZB-PKGsynth-GFP. Both the gBlock and the vector were digested with AgeI and AvrII and fragments were ligated resulting in vectors pZB-T695A-GFP and pZB-T695E-GFP.

All of the above-described plasmids, were linearised overnight with HindIII and transfected together with the pDC2-pkg Cas9 plasmid into *P. falciparum* line *pfpkg_2lox*. Correct integration was confirmed as described above and presence of mutations was confirmed by Sanger sequencing.

Construct pDC_loxnPKG:lox22mCherry [13] was substantially modified to create line *pkgS576A_S817A*. Initially, S576 of *pfpkg* in that plasmid was mutated to Alanine by overlapping PCR using primer sets St3_For/S576A_Rev, S576A_For/St4_Rev and St3_For/St4_Rev subsequently. The final PCR product was digested with HpaI/AgeI and ligated to vector pDC_loxnPKG:lox22mCherry previously digested with the same enzymes resulting in vector pDC_S576A:lox22mCherry. To add a second fragment of *pfpkg* harbouring mutation S817A, a differentially recodonised *pfpkg* fragment with the mutation, flanked at the 5′ by a SERA2 intron that contained a lox2272 site and at the 3′ by the N-terminus of mCherry was commercially obtained as a vector from GeneArt (ThermoFischer Scientific). Vector was digested with HpaI/MscI to isolate the fragment, which was cloned into vector pDC_mCherry_MCS [3] previously digested with the same enzymes. The resulting plasmid was digested with AleIv2/NheI and the released fragment was ligated to vector pDC_S576A:lox22mCherry, resulting in vector pDC_S576A:S817A. Correct sequence was verified by Nanopore sequence. Final plasmid was linearised with SacII and transfected to line *pfpkg_2lox* alongside plasmid pDC2-pkg.

For line *pkgS817A/S819A_S819A*, a fragment of synthetic *pfpkg* harbouring the S817A and S819A mutations was commercially obtained as a gBlock from IDT. This was cloned to vector pDC_S576A:S817A, resulting in vector pDC_S817A/S819A:S817A. To introduce the second mutation a fragment was commercially obtained (IDT) harbouring the S819A mutation. This was initially cloned in vector pDC_S576A:S817A using the AcII and BssSI sites. This intermediate vector was digested with HpaI/MscI to isolate the mutated fragment, which was subsequently cloned in vector pDC-mCherry-MCS [3]. This vector was excised by NheI/AleI digestion and inserted in vector pDC_S817A/S819A:S817A, previously digested with the same enzymes resulting in plasmid pDC_S817A/S819A:S819A. This plasmid was linearised with SacII and transfected to line *pfpkg_2lox* alongside plasmid pDC2-pkg.

## Parasite growth assays

Synchronous ring-stage parasites at 0.1% parasitaemia and 2% haematocrit were dispensed into 12-well plates. 50 μL samples from each well were collected at 0, 48, 96 and 144 hours. Samples were stained with SYBR green (ThermoFischer—1/10000 dilution from stock) and

analysed by flow cytometry on a BD FACSVerse using BD FACSuite software. For growth assays in the presence of C2, synchronous ring-stage parasites at 1% parasitaemia and 2% haematocrit were dispensed into 12-well plates. Samples were collected at 0 and 48h, stained with SYBR green and analysed by flow cytometry on a BD FACSVerse using BD FACSuite software. Data were analysed using FlowJo software.

## Parasite egress assay

Mature schizonts were isolated by Percoll centrifugation and incubated for a further 3 h in medium containing C2 (1 μM). After removal of the inhibitor, schizonts were immediately resuspended in fresh serum-free RPMI at 37°C to allow egress. Schizont pellets and culture supernatants at t = 0 were collected as control samples, whilst culture supernatants were collected by centrifugation after 60min.

## Immunoblotting and Immunofluorescence analysis

Synchronised schizonts were isolated by Percoll gradient centrifugation and washed in RPMI 1640 without Albumax. Parasite proteins were extracted with a Triton X-100 containing buffer (20 mM Tris-HCl pH 7.4, 150 mM NaCl, 0.5 mM EDTA, 1% Triton X-100), supplemented with cOmplete EDTA-free protease inhibitors (Roche) as previously described [13]. Supernatants were mixed with reducing SDS sample buffer and incubated for 5 min at 95°C prior to fractionation by SDS-PAGE analysis on 4–15% Mini-PROTEAN TGX Stain-Free Protein Gels (Bio-Rad). Western Blot analysis was performed as described previously [15].

For AMA1 relocalisation studies, vehicle (DMSO) or RAP-treated Percoll-enriched schizonts were incubated for 4 h in the presence of C2 and then in the presence of E64d (50 μM) for 45 min. Thin blood smears were fixed in 4% (w/v) paraformaldehyde and permeabilised in 0.1% (v/v) Triton X-100. After blocking with 3% BSA, smears were probed with a rabbit anti-AMA1, 1:200. Slides were probed with an Alexa Fluor 594-conjugated secondary antibody (1:2000). Images were collected using a Nikon Eclipse Ni-E wide field microscope with a Hamamatsu C11440 digital camera and 100x/1.45NA oil immersion objective. Identical exposure conditions were used at each wavelength for each pair of mock- and RAP-treated samples under comparison. Images were processed using Fiji software [55].

## Time-lapse and live fluorescence microscopy

Viewing chambers were constructed as previously described [15]. Images were recorded on a Nikon Eclipse Ni light microscope fitted with a Hamamatsu C11440 digital camera and Nikon N Plan Apo λ 63x/1.45NA oil immersion objective. For time-lapse video microscopy, differential inference contrast (DIC) images were taken at 10 sec intervals over 30 min while fluorescence images were taken every 2 min to prevent bleaching. Time-lapse videos were analysed and annotated using Fiji.

## Expression and purification of recombinant PfPKG proteins

Native coding sequences for wild-type PfPKG and mutant $PfPKG_{Y214A}$ were amplified from plasmids pTrc-PfPKG [4] and $pTrc-PfPKG_{Y214A}$ respectively (Phusion High-Fidelity DNA Polymerase, Thermo Scientific), using primer pair SH003 and SH004 (S1 Table). The full-length CDS for quintuple mutant $PfPKG_{STmut}$ and partial CDS for single mutants $PfPKG_{T695A}$ and $PfPKG_{T695E}$ were synthesised as gBlocks (Integrated DNA Technologies) using native codon usage. All but $PfPKG_{T695A}$ encoding DNA fragments were cloned into the XmaI/XhoI site of pET-47b (Novagen), allowing for expression of recombinant PfPKG with an N-terminal

hexa-histidine affinity tag in *E. coli* expression strains. For both PfPKG$_{T695A}$ and PfPKG$_{T695E}$ the corresponding wild-type sequence in the pET-47b PfPKG$_{WT}$ plasmid was removed with SnaBI/XhoI and replaced with the sequence containing the appropriate mutation. Optimal protein expression and solubility was achieved in Rosetta (DE3) pLysS (Novagen) for wild-type His::PfPKG, mutant His::PfPKG$_{Y214A}$, His:PfPKG$_{T695A}$, His:PfPKG$_{T695E}$ and in OverExpress C43(DE3) cells (Sigma-Aldrich) for the quintuple mutant His::PfPKG$_{STmut}$, respectively. Bacterial cultures were grown at 37˚C in Terrific Broth (containing the appropriate antibiotics) until an OD$_{600}$ of ~ 0.6–0.8 and protein expression was induced with 1 mM IPTG for 16–18 h at 18˚C.

Bacterial cells were harvested by centrifugation at 4,000 xg (30 min, 4˚C), resuspended in IMAC buffer A [50 mM Tris-Cl pH 7.4, 300 mM NaCl, 10 mM imidazole, 5% glycerol, 1 mM DTT] supplemented with 5 mM MgCl$_2$ and DNaseI (1 mg/ml) and lysed using a high-pressure homogeniser (EmulsiFlex, Avestin). Following solubilisation with 1% Triton X-100 for 30 min at 4˚C, the lysate was cleared at 40,000 xg (45 min, 4˚C), passed through a 0.45 μm syringe filter and loaded onto an equilibrated HisTRAP HP column (GE Healthcare) for ÄKTA-assisted immobilised metal affinity chromatography (IMAC). Protein-bound columns were washed with ten column volumes of IMAC buffer A containing 50 mM imidazole. Recombinant proteins were eluted in a linear imidazole gradient using IMAC buffer B [IMAC buffer A containing 0.5 M imidazole]. Fractions containing the protein of interest were concentrated (Amicon Ultracel 30 K, Millipore) and further purified by size-exclusion chromatography (SEC) using a Superdex S200 10/300 GL column (GE Healthcare) in SEC buffer [25 mM HEPES pH 7.4, 150 mM NaCl, 5% glycerol, 2 mM DTT]. All buffers contained cOmplete EDTA-free protease inhibitors (Roche). Purity of the recombinant PfPKG proteins was assessed by SDS-PAGE using NuPAGE Novex 3–8% Tris-Acetate gels (Invitrogen) and expression was confirmed by Western blot analysis.

## ParM-ADP kinase assays

The ADP-specific biosensor MDCC-ParM [33] was utilized to compare the enzymatic activities and determine the kinetic parameters of recombinant wild-type and mutant PfPKG in real-time measurements of kinase activity. MDCC-ParM was prepared as described previously [33]. Phosphorylation of the substrate PKAtide (GRTGRRNSI) by recombinant PfPKG was monitored by detecting the generation of ADP upon PfPKG-dependent ATP hydrolysis and binding of ADP to the coumarin-labelled biosensor.

To establish initial velocities and optimal enzyme concentrations for downstream kinetic studies, kinase assays were carried out at various protein concentrations in reaction buffer [25 mM HEPES, 25 mM KCl, 5 mM MgCl$_2$, 0.1 mg/ml BSA, 2 mM DTT, pH 7.4] containing 10 μM MDCC-ParM, 10 μM cGMP and 4 μM PKAtide. Reactions were initiated by addition of 20 μM ATP (ATP disodium salt ≥98%, Ultra Pure Grade, Sigma) and allowed to proceed for up to 50 min at 25˚C. To determine the activation constant K$_A$ for cGMP and Michaelis-Menten constant K$_m$ for PKAtide, respectively, cyclic nucleotide and substrate concentrations were varied by serial dilutions and enzymes used at ~ 0.2 U. MDCC fluorescence was measured in a CLARIOstar microplate reader (BMG Labtech) at excitation and emission wavelengths of 439 and 474 nm, respectively. An ADP/ATP-calibration curve was established as previously described to calculate enzyme velocities from the slopes obtained by linear regression [33]. K$_A$ and K$_m$ values were obtained by using the non-linear regression fit 'Specific binding with Hill slope' and 'Michaelis-Menten', respectively (https://www.graphpad.com). Enzyme velocities were normalised as described in [56] to account for protein batch and experimental variability.

## Circular dichroism (CD) analysis

Recombinant PfPKG was dialysed into PBS (pH 7.4) over night at 4°C and adjusted to 0.15 mg/ml for data collection using a JASCO J-815 CD spectropolarimeter. 140 µl of protein sample was measured using a 1 mm quartz cuvette (Hellma Analytics). Far-UV CD spectra were recorded between 260 and 190 nm with the following parameters: 200 nm/min scanning speed, 0.25 sec DIT, 0.2 nm data pitch and 2 nm bandwidth at 25°C. Raw data were corrected by subtraction of buffer/blank spectra and converted to $\Delta\epsilon$ on a mean residue weight basis using in-house software [57]. Secondary structure content was estimated using methods originally described by Sreerama and Woody [58] in CDPro software.

## Kinase-inhibition assays and $K_m$ determination for ATP

ADP-Glo (Promega) kinase assays were carried out to examine the effect of inhibitors ML-10 and C2 on the activity of recombinant PfPKG proteins. End-point assays were performed with 1 nM recombinant protein in assay buffer [25 mM HEPES, 25 mM KCl, 5 mM $MgCl_2$, 0.1 mg/ml BSA, 2 mM DTT, pH 7.4]. Kinase reactions were initiated by adding the substrate reaction mix containing 25 µM PKAtide, 10 µM cGMP and 10 µM ATP in assay buffer +/- inhibitor and were allowed to proceed for 60 min at 25°C. ADP to ATP conversion and luciferase reaction were undertaken as per manufacturer's recommendation and luminescence was recorded on a SpectraMax M5$^e$ (Molecular Devices). RLUs were corrected for baseline luminescence measured in the absence of any protein.

ADP-Glo (Promega) kinase assays were also used to determine the $K_m$ for ATP. ATP concentrations (0–800 µM) were varied by serial dilutions in assay buffer containing 50 µM PKAtide and 10 µM cGMP and the reaction was allowed to proceed for different timepoints (0, 5, 10, 20, 30, 40 and 60 min) per concentration to establish initial velocities. ADP to ATP conversion and luciferase reaction were performed as described above.

## IP and MS analysis for *P. falciparum* schizonts

For pull-down analysis, frozen schizont preparations from lines PfPKG-HA, PKG-GFP, *pkgT695A*, *pkgT695E* and B11 were resuspended in 100 µl of lysis buffer [20 mM tris-HCl (pH 7.5), 150 mM NaCl, 1 mM EDTA, and 1% Triton X-100, complete protease inhibitors and PhosSTOP] and extracted at 4°C for 30 min with intermittent vortexing. Extracts were clarified by centrifugation at 16,000g for 20 min at 4°C, filtered through 0.22 Corning Costar Spin-X centrifuge tube filters, and then incubated with 25 µl of anti-HA magnetic beads (Pierce) or with 25 µl of GFP-Trap Magnetic Agarose (ChromoTek) overnight at 4°C. All samples were processed following manufacturer's instructions.

**T695 phosphorylation.** Bands corresponding to the molecular weight region of PKG_HA protein were excised from the gel and washed. Reduced and alkylated proteins were in-gel digested with 100 ng of trypsin (modified sequencing grade, Promega) overnight at 37°C. Supernatants were dried in a vacuum centrifuge and resuspended in 0.1% trifluoroacetic acid (TFA). Liquid chromatography electrospray ionization tandem mass spectrometry (LC-ESI-MS/MS) was performed on an Orbitrap Fusion Lumos Tribrid Mass Spectrometer (Thermo Fisher Scientific) equipped with an EasySpray source and UltiMate 3000 RSLCnano UHPLC liquid chromatography system (Thermo Fisher Scientific). The digested protein from each band was injected in triplicate. 7 µl per injection was loaded onto a 2 mm by 0.3 mm Acclaim PepMap C18 trap column (Thermo Fisher Scientific) in 0.1% TFA at 15 µl/min before the trap being switched to elute at 0.25 µl/min through a 50 cm by 75 µm EASY-Spray C18 column. A ~90′ gradient of 2 to 20% B over 57′ and then 20 to 40% B over 25′ was used followed by a short gradient to 90% B held for 7' and back down to 2% B and a 20′ equilibration in 2% B

[A = 2% acetonitrile (ACN), 5% DMSO, 0.1% formic acid (FA); B = 75% ACN, 5% DMSO 0.1% FA]. The Orbitrap Fusion Lumos was operated in "Data Dependent Acquisition" mode with the MS1 full scan at a resolution of 120,000 FWHM, followed by as many subsequent MS2 scans on selected precursors as possible within a 3-s maximum cycle time. MS1 was performed in the Orbitrap with an AGC target of $4 \times 105$, a maximum injection time of 50 ms, and a scan range from 375 to 1500 m/z. MS2 was performed in the ion trap with a rapid scan rate, an AGC target of $2 \times 103$, and a maximum injection time of 300 ms. Isolation window was set at 1.2 m/z, and 38% normalized collision energy was used for HCD. Dynamic exclusion was used with a time window of 40 s.

Peak lists (MGF file format) were generated from raw data and searched against a custom database of P. falciparum schizont proteins using Mascot (Matrix Science, London, UK). Trypsin was selected as the enzyme. Precursor ion tolerance was set to 10 parts per million (ppm) and fragment ion tolerance 0.6 Da. Variable amino acid modifications were oxidized methionine, protein N-term acetylation, and phosphorylated serine, threonine, and tyrosine. Fixed amino acid modification was carbamidomethyl cysteine. The results files were downloaded from Mascot and imported as a peptide search in Skyline [59] to generate a spectral library using the DDA with MS1 filtering workflow to measure quantitative differences in peptide expression using the MS1 extracted ion chromatograms.

**PKG interactome.** Protein complexes were eluted from beads with 80 uL reduced 2× SDS sample buffer. Samples were incubated at 95°C for 5 min and run on 12% SDS-PAGE gels. Samples were run 10mm from the top on the SDS-PAGE gel. Gel bands were then excised and placed into separate protein LoBind Eppendorf tubes, de-stained with 50/50, 50 mM ammonium bicarbonate (AmBic)/acetonitrile (ACN). Gel bands were then dehydrated with 100% ACN followed by reduction with dithiothreitol (10 mM DTT) at 37°C for 30 min, then alkylated with iodoacetamide (55 mM IAM) for 20min in the dark. Finally rehydrated with a 50 mM AmBic, containing 100 ng of Trypsin (Pierce Trypsin Protease, MS Grade), and incubated at 37°C overnight. The Recovered peptides were dried by vacuum centrifugation then re-solubilised in 0.1% formic acid prior to for LC-MS analysis.

**Mass spectrometry data acquisition.** The tryptic peptides were analyzed on a Thermo-Fisher Scientific Fusion Lumos mass spectrometer (Lumos Tribrid Orbitrap mass spectrometer, ThermoScientific, San Jose, USA) coupled to an UltiMate 3000 HPLC system for on-line liquid chromatographic separation. Tryptic peptides for each sample were loaded onto a C18 trap column (ThermoFisher Scientific Acclaim PepMap 100; 75 µM × 2 cm) then transferred onto a C18 reversed-phase column (ThermoFisher Scientific PepMap RSLC; 50 cm length, 75 µm inner diameter). Peptides were eluted with a 90 min gradient at flow rate of 250 nL/min. The following gradients were used: 5% B for 0–7 min, 8–55% B from 7 to 68 min, 95%B from 70 to 74 min, and re-equilibration at 5% B. (buffer A: 5% (v/v) DMSO, 95% (v/v) 0.1% formic acid in water; B: 5% (v/v) DMSO, 20% (v/v) 0.1% formic acid in water, 75% (v/v) 0.1% formic acid in acetonitrile). The analytical column was directly interfaced, via a nano-flow electrospray ionisation source, to the mass spectrometer. The instrument was operated in "Data Dependent Acquisition" mode using a resolution of 120,000 for the full MS from m/z 375–1500. This was followed by MS/MS using 32% high energy collision dissociation (HCD). Dynamic exclusion was used with a time window of 30s.

**Data analysis.** Raw Data files were processed on MaxQuant software (version 2.0.3.0). The LFQ algorithm and match between runs settings were selected. Enzyme specificity for trypsin was selected (cleavage at the C-terminal side of lysine and arginine amino acid residues unless proline is present on the carboxyl side of the cleavage site) and a maximum of two missed cleavages were allowed. Cysteine carbamidomethylation was set as a fixed modification, while oxidation of methionine and acetylation of protein N-termini were set as variable

modifications. Data were searched against PlasmoDB-63. MaxQuant also searched the same database with reversed sequences of 1% false discovery rate at peptide and protein levels. A built-in database of common protein contaminants was also searched. The 'proteingroups.txt' output file generated on MaxQuant was loaded in Perseus version 1.4.0.2. Contaminant and reverse protein hits were removed. LFQ intensities were log2 transformed. For each sample, the triplicate was grouped. Data were filtered for at least two out of the three replicates LFQ intensity values in at least one group. Protein LFQ intensities were normalised, and Missing values (NaN) were imputed from a normal distribution with default values. A protein was considered significantly differentially expressed when FDR < 0.05.

## *P. falciparum* phosphoproteomics

Parasite pellets from lines PfPKG_GFP, and RAP treated pellets from lines *pkgT695A* and *pkgT695E* were treated as previously described [15]. In brief, tightly synchronised and highly segmented Percoll-enriched schizonts were treated for 2 h with C2 and then washed to allow initiation of the egress cascade for 15 minutes. Schizonts were then lysed in the presence of 0.15% saponin in PBS containing cOmplete Mini EDTA-free Protease and PhosSTOP Phosphatase inhibitor cocktails (Roche) for 10 min. Pellets were washed twice in PBS and snap frozen in liquid nitrogen. Pellets were resuspended in 1 mL 8 M urea in 50 mM HEPES, pH 8.5, containing protease and phosphatase inhibitors and 100 U/mL benzonase (Sigma). Proteins were extracted from the pellets by sonication. Samples were incubated on ice for 10 min and centrifuged for 30 min at 14,000 rpm at 4°C. In total 3 biological replicates and 2 technical replicates were obtained from each line (n = 18 samples in total). Protein content was estimated by a BCA protein assay (Thermo Scientific) and 200 μg of each sample were reduced with 10 mM dithiothreitol for 60 min at room temperature and then alkylated with 20 mM iodoacetamide for 30 min at room temperature. Samples were digested initially with LysC (WAKO) at 37°C for 2.5 h and subsequently with 10 μg trypsin (modified sequencing grade, Pierce) overnight. After acidification, $C_{18}$ Macro-Spin columns (Nest Group) were used to clean up the digested peptide solutions and the eluted peptides dried by vacuum centrifugation. Samples were resuspended in 50 mM HEPES and labelled using the 0.5 mg TMTpro 18-plex Label Reagent Set (Thermo Scientific) following manufacturer's instructions. The eluted TMT-labelled peptides were dried by vacuum centrifugation and phosphopeptide enrichment was subsequently carried out using the sequential metal oxide affinity chromatography (SMOAC) strategy with High Select $TiO_2$ and Fe-NTA enrichment kits (Thermo Scientific). Eluates were combined prior to fractionation with the Pierce High pH Reversed-Phase Peptide Fractionation kit (Thermo Scientific). The dried TMT-labelled phospho-peptide fractions generated were resuspended in 0.1% TFA for LC-MS/MS analysis using a U3000 RSLCnano system (Thermo Scientific) interfaced with an Orbitrap Fusion Lumos (Thermo Scientific). Each peptide fraction was pre-concentrated on an Acclaim PepMap 100 trapping column before separation on a 50-cm, 75-μm I.D. EASY-Spray Pepmap column over a three hour gradient run at 40°C, eluted directly into the mass spectrometer. The instrument was run in data-dependent acquisition mode with the most abundant peptides selected for MS/MS fragmentation. Two replicate injections were made for each fraction with different fragmentation methods modified for the TMTpro reagents [60]. The acquired raw mass spectrometric data were processed in MaxQuant (v 2.4.2.0) and searched against PlasmoDB-63 (https://plasmodb.org). Fixed modifications were set as Carbamidomethyl (C) and variable modifications set as Oxidation (M) and Phospho (STY). The estimated false discovery rate was set to 1% at the peptide, protein, and site levels. A maximum of two missed cleavages were allowed. Reporter ion $MS^2$ or Reporter ion $MS^3$ was appropriately selected for each raw file. Other parameters were used as preset in the software. The MaxQuant output file PhosphoSTY Sites.txt, an FDR-controlled site-

based table compiled by MaxQuant from the relevant information about the identified peptides, was imported into Perseus (v1.4.0.2) for data evaluation.

For a phosphorylation site to be considered regulated, the following cut-offs were applied: $P$-value < 0.05, Welch difference > 1 or < -1 (2-fold) and localisation probability >0.7. Motif identification was performed using the MEME tool [61]. Significantly hyper and hypo-phosphorylated proteins were used for enrichment of GO terms. GO IDs were extracted from the *P. falciparum* annotation file (PlasmoDB.org). Enriched GO terms were identified as described before [62]. Plots were made using R package ggplot2 (https://ggplot2.tidyverse.org). Deregulated proteins from each Phosphoproteome were imported to STRING database (version 12.0) using a confidence threshold of >0.4. The interaction networks were visualised in Gephi (https://gephi.org) using the Fruchterman Reingold layout. Significantly deregulated sites were imported to Cytoscape v 3.10.1 and mapped using the Omics Visualizer App [63,64].

## Peptide arrays

Oriented Peptide Array Library (OPAL) synthesis was performed by the Francis Crick Institute Peptide Chemistry Science Technology platform. Briefly, peptide arrays were synthesised on an Intavis ResResSL automated peptide synthesiser (Intavis Bioanalytical Instruments, Germany) by cycles of N(a)-Fmoc amino acids coupling via activation of the carboxylic acid groups with diisopropylcarbodiimide in the presence of ethylciano-(hydroxyamino)-acetate (Oxyma pure) followed by removal of the temporary a-amino protecting group by piperidine treatment. Side chain protection groups were subsequently removed by treatment of membranes with a deprotection cocktail (20 ml 95% trifluoroacetic acid, 3% triisopropylsilane and 2% $H_2O$) for 4 h at room temperature, then washing (4x dichloromethane, 4x ethanol, 2x $H_2O$ and 1x ethanol) prior to being air dried. The final product is a cellulose membrane containing a library of 9-mer peptides with the general sequences: A-X-X-X-X-S-X-X-X-X-A or A-X-X-X-X-T-X-X-X-X-A. For each peptide, one of the 20 naturally occurring proteogenic amino acids was fixed at each of the 8 positions surrounding the phosphorylated residue (S or T), with the remaining positions, represented by X, degenerate (approximately equimolar amount of the 16 amino acids excluding cysteine, serine, threonine and tyrosine). Cellulose membranes were placed in an incubation trough and moistened with 2 ml ethanol. They were subsequently washed twice with 50 ml kinase buffer (25 mM HEPES, 10 mM magnesium chloride pH 7.4) and incubated overnight in 100 ml reaction buffer (kinase buffer + 0.2 mg/ml BSA (BSA Fraction V, Sigma) + 50 µg/ml kanamycin). The following day, membranes were incubated at 30°C for 1 h in 30 ml blocking buffer (kinase buffer + 1mg/ml BSA + 50µg/ml kanamycin). After incubation, the blocking buffer was replaced with 30ml reaction buffer supplemented with 25 µM ATP (for PfPKG$_{WT}$) or 45 µM ATP (for PfPKG$_{T695A}$ and PfPKG$_{T695E}$), 10 µM cGMP and 125µCi [g-32P]-ATP (Hartmann Analytics, Germany). The reaction was started by adding 50 nM of the recombinant PKG and left to incubate for 20 min at 30°C with gentle agitation. After incubation, the reaction buffer was removed and the membranes were washed 10 x 15 min with 1 M NaCl, 3 x 5 min with H2O, 3 x 15 min with 5% H3PO4, 3 x 5 min with H2O and 2 x 2 min with ethanol. Membranes were then left to air dry before being wrapped in plastic film and exposed overnight to a PhosphorScreen. The radioactivity incorporated into each peptide was then determined using a Typhoon FLA 9500 phosphorimager (GE Healthcare) and quantified with the program ImageQuant (version 8.2, Cytiva Life-Science). Data corresponding to the "signal above background" were used.

A scoring matrix was computed as previously described [65]. The resulting PSSMs based on each OPAL data set were used to produce sequence logos with Seg2Logo 2.0 [66]. Heatmaps were made in R studio using package ggplot2 (https://ggplot2.tidyverse.org).

## Statistical analysis

All statistical analysis was carried out using GraphPad Prism9.

## Supporting information

**S1 Fig. Phosphosite mutations destabilise the pentagonal architecture of PfPKG. (A)** Cartoon representation of the *P. falciparum* PKG x-ray crystal structure (PDB ID: 5DYK) in its apo form where predicted phosphosites are indicated and shown as sticks within colour matching spheres **(B)** Schematic representation of the approach used to create line *pkgsynth_GFP*. Blue line indicates the position targeted by the gRNA. RAP-induced DiCre activity switches expression from wt PKG to a gene replacement with a partially synthetic *pfpkg* gene fused to eGFP. Black arrows; oligonucleotides used for excision PCR. **(C)** Excision PCR showing generation of products after activation of DiCre upon RAP treatment in line *pkgsynth_GFP* and *pkgΔPhos*. Expected sizes of the amplicons corresponding to non-excised and excised fragments are 5kb and 1.4 kb respectively. **(D)** Growth curves showing replication of DMSO-treated (control) or RAP-treated *pkgsynth_GFP* and *pkgΔPhos* lines. Mean values are shown. Error bars: ± SD (n = 2). **(E)** Representative still images of a 30 min time-lapse videoof mock-treated (grey) or RAP-treated (green) parasites of lines *pkgsynth_GFP* and *pkgΔPhos*. *pkgsynth_GFP* RAP-treated schizonts underwent egress (upper panel), whilst *pkgΔPhos* RAP-treated parasites did not (lower panel). Scale bar, 10 μM. **(F)** (Left panel) Western blot showing expression of PKG (expected MW: 98 kDa) in DMSO-treated *pkgsynth_GFP* schizonts relevant to the parental line (WT). Upon RAP-treatment there is appearance of a signal corresponding to the PKG_GFP fusion (expected molecular weight: 125 kDa). (Right panel) Representative Western blot of *pkgΔPhos* schizonts after DMSO or RAP treatment. Note the multiple bands appearing at the RAP-treated sample. Cytoplasmic HSP-70 was used as a loading control. **(G)** Cartoon representation of PfPKG (PDB: 5DYK) with Y694 in green, the CNB-A to CNB-B connecting helix in a turquoise-blue gradient and the PfPKG kinase activation loop residues in white. (Left panel) The wild type PKG Y694 is engaged in stabilizing interactions with the helix residues K157 and S160. In the PKG mutant Y694A, both interactions are lost, resulting in local structural destabilization of the protein (Right panel). Images were generated in PyMOL.
(TIF)

**S2 Fig. Generation and genotyping of transgenic P. falciparum lines. (A-B)** Modification strategies and genotyping data for generation of the parasite lines *pkgΔPhos*, *pkgY214A*, *pkgY214F*, *pkgT202A*, *pkgF96A*, *pkgSTmut*, *pkgT695A* and *pkgT695E* respectively. Blue line indicates the position targeted by the gRNA. Relative positions of mutations are depicted by coloured stars. Positions of oligonucleotides used for genotyping by diagnostic PCR are indicated (coloured arrows), and agarose gel electrophoresis of corresponding PCR products are shown. Positions of *lox* sites are indicated with coloured arrowheads (yellow, *loxN*; green, *lox2272*). **(C)** Modification strategy for lines *pkgS576A_S817A* and *pkgS817A/S819A_S819A*. Blue line indicates the position targeted by the gRNA. Relative positions of mutations are depicted by coloured stars. Positions of oligonucleotides used for genotyping by diagnostic PCR are indicated (coloured arrows), and agarose gel electrophoresis of corresponding PCR products are shown. Positions of *lox* sites are indicated with coloured arrowheads (yellow, *loxN*; green, *lox2272*).
(TIF)

**S3 Fig. F96 and T202 mutations do not affect PfPKG function in vivo. (A)** Schematic representation of the approach used to create lines *pkgF96A* and *pkgT202A*. Relative positions of the

mutated amino acids are denoted by blue stars. **(B)** Growth curve showing replication of DMSO-treated (control) or RAP-treated *pkgF96A*. Mean values are shown. Error bars: ± SD (n = 3). **(C)** Western blot showing expression of endogenous PKG in DMSO-treated *pkgT202A* schizonts and of the mutated PKG (T202A) fused to GFP upon RAP-treatment. **(D)** Growth curve showing replication of DMSO-treated (control) or RAP-treated *pkgT202A*. Mean values are shown. Error bars: ± SD (n = 3).
(TIF)

**S4 Fig. Recombinant expression and kinetic properties of the different PfPKG mutants.**
**(A)** Coomassie stained gels and western blot analysis of the 5 different recombinant PfPKGs used in this study. **(B)** Determination of PKAtide $K_m$, cGMP $K_A$ and ATP $K_m$ for $PfPKG_{WT}$, $PfPKG_{Y214A}$, $PfPKG_{STmut}$, $PfPKG_{T695A}$. **(C)** Summary of kinetic parameters, shown are the means ± SEM and Two-way ANOVA. **(D)** $PfPKG_{T695A}$ and $PfPKG_{T695E}$ activity assay showing phosphorylation of PKAtide.
(TIF)

**S5 Fig. Phosphomutants of the N and C-lobe do not affect parasite viability. (A)** Schematic of line *pkgS576A_S817A*. RAP-induced DiCre activity switches expression from wt PKG to either a gene replacement with a S576A mutant fused to eGFP (recombination event 1; PKG_S576A) or to expression of a S817A mutant fused to mCherry (recombination event 2; PKG_S817A). Similar allelic replacement strategy applies to line *pkgS817A/S819A_S819A*. Black and red arrows; oligonucleotides used for identification of both events by diagnostic PCR. **(B)** Representative images from DIC/fluorescence microscopic examination of RAP-treated *pkgS576A_S817A* and RAP-treated *pkgS817A/S819A_S819A* parasites (end of cycle 0), showing both GFP-and mCherry-positive schizonts. Scale bar, 10 μm. **(C)** Stills from time-lapse DIC/fluorescence microscopy of isolated, RAP-treated *pkgS576A_S817A* and *pkgS817A/S819A_S819A* schizonts, showing that all four mutants can undergo egress. Scale bars, 10μm.
(TIF)

**S6 Fig. The PKA residue network that stabilizes the phosphate group in the phosphorylation site is not fully conserved in PKG.** The molecular structures of PKG (light grey sticks coloured by elements, PDB: 5dyk) and PKA (mustard sticks coloured by elements, PDB: 1atp) were superimposed (rmsd: 1.3Å). The structurally equivalent residue side chains from the phosphorylation site are depicted here and labelled. Hydrogen bonds in the phosphorylated PKA are seen as black dashed lines with the phosphate group in orange. Structurally stabilizing PKA residues H87 (in the C-helix) and T195 (in the Activation loop) are not conserved in PKG (N585 and A693 respectively).
(TIF)

**S7 Fig. T695 is phosphorylated from trophozoite stages, prior to PKG activation. (A)** Western blot and Coomassie stained gel after PKG pulldown showing presence of the protein in both trophozoites (34h) and schizonts (46h). **(B)** Peak areas of precursor ion chromatograms are extracted after MS1 filtering for peaks picked based on MS2 peptide identification in three technical replicates at both trophozoite (T1-T3) and schizont samples (S1_S3). Peaks for the phosphorylated T695 peptide were found in all experiments (Left panel) showing similar ratio between trophozoite and schizont samples as the control peptide (Right panel).
(TIF)

**S8 Fig. Protein-protein interaction networks of T695A and T695E deregulated proteins.**
**(A)** STRING network (y Files Hierarchical layout) visualizing functional interactions (edges) between proteins (nodes) significantly dephosphorylated in the T695A mutant over WT.

Default STRING clustering confidence score cutoff of 0.4 was used to determine whether two nodes were functionally related. Single proteins are excluded from the schematic. Significantly deregulated phosphosites are depicted on each protein and were coloured according to the Welch difference (outer visualisation) and the multiplicity (number of phosphorylation events in the peptide–inner visualisation). Names or PlasmoDB IDs are displayed on each protein. **(B)** STRING network (y Files Hierarchical layout) visualizing functional interactions (edges) between proteins (nodes) significantly dephosphorylated in the T695E mutant over WT. Same clustering parameters and visualisation were used as in (A).
(TIF)

**S9 Fig. Gene ontology analysis of T695A and T695E phosphoproteomes Gene ontology (GO) enrichment analysis of significantly hyper or hypo phosphorylated proteins in T695A and T695E schizonts.** GO terms were obtained from the PlasmoDB database. Size of the bubble indicates the level of significance (1-p value) of the enriched GO term and colour density indicates the number of differentially expressed proteins (log2 protein count) associated with the GO term.
(TIF)

**S10 Fig. OPAL activity assays and determined sequence logos.** Autoradiographs of the oriented peptide array libraries showing phosphorylation activity of PfPKGWT, PfPKGT695A and PfPKGT695E. Library was based on the design A-X-X-X-X-S-X-X-X-X-A or A-X-X-X-X-T-X-X-X-X-A.
(TIF)

**S1 Movie. Time-lapse video microscopy of DMSO- and RAP-treated pkgsynth_GFP.** Both, DMSO-treated parasites (DIC) and RAP-treated parasites (expressing eGFP) egress normally.
(MP4)

**S2 Movie. Time-lapse video microscopy of DMSO- and RAP-treated ΔPhos parasites.** DMSO-treated parasites egress normally (DIC) whilst RAP-treated ΔPhos schizonts (eGFP) cannot undergo egress.
(MP4)

**S3 Movie. Time lapse video microscopy of RAP-treated Y214A parasites (expressing eGFP) failing to egress compared to DMSO-treated ones (DIC).**
(MP4)

**S4 Movie. Time-lapse video microscopy of RAP-treated pkgS576A_S817A parasites.** Both the green (S576A_eGFP) and the red (S817A_mCherry) schizonts undergo egress.
(MP4)

**S5 Movie. Time-lapse video microscopy of RAP-treated pkgS817A/S819A_S819A parasites.** Both the green (S817A/S819A_eGFP) and the red (S819A_mCherry) schizonts undergo egress.
(MP4)

**S6 Movie. Time-lapse video microscopy of DMSO- and RAP-treated T695A parasites.** DMSO-treated parasites egress normally (DIC) whilst RAP-treated T695A schizonts (eGFP) cannot undergo egress.
(MP4)

**S7 Movie. Time-lapse video microscopy of DMSO- and RAP-treated T695E parasites.** DMSO-treated parasites egress normally (DIC) whilst RAP-treated T695E schizonts (eGFP)

cannot undergo egress.
(MP4)

**S1 Data. Kinetic properties of the recombinantly expressed PfPKGWT, PfPKGY214, PfPKGSTmut, PfPKGT695A, PfPKGT695E and CD analysis of PfPKGWT and PfPKGSTmut.**
(XLSX)

**S2 Data. Proteins identified from pull down experiments of WT, PKG_GFP, PKG_T695A and PKG_T695E.**
(XLSX)

**S3 Data. Peak areas from the T695 phosphorylation.**
(XLSX)

**S4 Data. Phosphoproteome analyses of PKG_GFP, PKG_T695A and PKG_T695E in *P. falciparum* schizonts.**
(XLSX)

**S1 Table. Oligonucleotide primers used in this study.**
(DOCX)

**S1 Raw Data. The source data file contains the raw data of graphs and charts not included in S1–S4 Data.**
(XLSX)

## Acknowledgments

We are grateful to Abhinay Ramaprasad (The Francis Crick Institute) and Hugo Belda (Gulbenkian Institute of Science) for their help on GO annotation analysis and OPAL assays, respectively. Authors would like to thank Steven Howell for assistance in generating the T695 phosphorylation dataset. We also thank Nicola O'Reilly and Dhira Joshi from the Chemical Biology Science Technology platform (The Francis Crick institute) for the PKAtide peptide and the OPAL libraries, respectively.

## Author Contributions

**Conceptualization:** Konstantinos Koussis, Silvia Haase, David A. Baker, Michael J. Blackman.

**Data curation:** Konstantinos Koussis, Silvia Haase, Chrislaine Withers-Martinez, Helen R. Flynn, Simone Kunzelmann, Evangelos Christodoulou, Fairouz Ibrahim.

**Formal analysis:** Konstantinos Koussis, Silvia Haase, Chrislaine Withers-Martinez, Helen R. Flynn, Simone Kunzelmann, Evangelos Christodoulou, Fairouz Ibrahim, Mark Skehel.

**Funding acquisition:** David A. Baker, Michael J. Blackman.

**Investigation:** Konstantinos Koussis, Silvia Haase, Chrislaine Withers-Martinez, Michael J. Blackman.

**Methodology:** Konstantinos Koussis, Chrislaine Withers-Martinez, Mark Skehel.

**Supervision:** Konstantinos Koussis, Simone Kunzelmann, Mark Skehel, David A. Baker, Michael J. Blackman.

**Validation:** Konstantinos Koussis, Silvia Haase, Chrislaine Withers-Martinez, Helen R. Flynn, Evangelos Christodoulou, Fairouz Ibrahim, Michael J. Blackman.

**Visualization:** Konstantinos Koussis, Silvia Haase, Chrislaine Withers-Martinez, Helen R. Flynn, Fairouz Ibrahim.

**Writing – original draft:** Konstantinos Koussis, Silvia Haase, David A. Baker, Michael J. Blackman.

**Writing – review & editing:** Konstantinos Koussis, Silvia Haase, Chrislaine Withers-Martinez, Helen R. Flynn, Simone Kunzelmann, Evangelos Christodoulou, Fairouz Ibrahim, Mark Skehel, David A. Baker, Michael J. Blackman.

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
