## [Decision Letter · Decision Letter 0]

3 May 2024

Dear Dr. Koussis,

Thank you very much for submitting your manuscript "Activation loop phosphorylation and cGMP saturation of PKG regulate egress of malaria parasites." for consideration at PLOS Pathogens. As with all papers reviewed by the journal, your manuscript was reviewed by members of the editorial board and by several independent reviewers. The reviewers appreciated the attention to an important topic. Based on the reviews, we are likely to accept this manuscript for publication, providing that you modify the manuscript according to the review recommendations.

As you will see in the reports, all reviewers are positive about the importance and relevance of the work. There are some concerns that should be addressed prior to publication. These concerns can be addressed with adjustments in the text, adding more details to experimental methods, and clarification on some experiments. IMPORTANT: While the experimental requests from reviewer #3 (double/triple activation loop mutants) will strengthen the study, we understand that these are not easy reagents to generate. Therefore, we do not require you to perform that experiment.

Sincerely,

Vasant Muralidharan

Guest Editor

PLOS Pathogens

Margaret Phillips

Section Editor

PLOS Pathogens

Michael Malim

Editor-in-Chief

PLOS Pathogens

orcid.org/0000-0002-7699-2064

Reviewer Comments (if any, and for reference):

Reviewer's Responses to Questions

**Part I - Summary**

Reviewer #1: Koussis et al. describe the importance of phosphorylation of the seven residues essential within the CNB-B and the kinase domain of PKG and how they regulate PfPKG function. The authors used a conditional allelic replacement induced by rapamycin approach as well as recombinant expression of several PfPKG mutants and the wild type. The authors show that reduced cGMP binding to cyclic nucleotide-binding domain B does not affect in vitro kinase activity but does prevent parasite egress. Also, they identified T695 as an essential residue in the activation loop, contributing to PKG kinase regulation. However, the activation loop phosphorylation is not the mechanism for kinase activation by itself. Overall, this study provides important new insights into how PfPKG is regulated and in my opininion is an excellent fit for PLoS Pathogens.

Reviewer #2: cGMP signaling is a critical regulator of a number of processes in apicomplexan biology, including invasion, motility, and egress. While most well-studied model organisms have multiple cGMP effectors, parasites such as Plasmodium have a single effector kinase. In the present study, Koussis and coauthors dissect the an unusual regulation phosphorylation mechanism of PfPKG. There was much to like about this study, it is clearly written and logically presented. Overall I found the figures easy to parse and the data convincing.

Reviewer #3: This is a very thorough and systematic study of the role of phosphorylation sites in the cGMP-dependent kinase of the malaria parasite Plasmodium falciparum. The reverse genetics experiments are complex but well conducted and well-controlled. Some of the observations are surprising, but the data are solid. In particular, the dispensability of the phosphorylation of the activation T-loop residue T695 differs from the usual role of this phosphorylation in many other kinases (including AGCs). This means that PfPKG is regulated in a fundamentally divergent way to mammalian PKGs (but see comments below).

**Part II – Major Issues: Key Experiments Required for Acceptance**

Reviewer #1: No major issues.

Reviewer #2: The authors appear to have missed an opportunity to combine their comparative changes in phosphoproteomics of their mutants with the in vitro OPAL peptide dataset. Do the subtle changes in the peptide dataset correlate with changes in the phosphoproteome (or are they at least mostly consistent?). If so, that would strengthen the paper. While it would be disappointing if that is not the case, it doesn't diminish the quality of the data, and would be important to explicitly note so alternative hypotheses can be considered by the field.

Reviewer #3: MAJOR POINT:

Overall, this is a solid study with unexpected but well-supported outcomes. I have only one major suggestion, that relates to the aforementioned unexpected finding:

P. 18 and Fig 4H. The observation that the Tyr695�Ala mutation does not affect kinase activity is surprising (in comparison to regulatory mechanisms in other AGC kinases where phosphorylation is a requisite for activity) but well supported by compelling data (Fig. 4H), Since there is an adjacent Tyrosine (Y694) or nearby Threonine (T699), phosphorylation of the latter may be required for activation. A double activation loop mutant (Y694-T695 or T695-T699, or a triple mutant), would be interesting to analyse.

In other kinases of the AGC group, activation loop phosphorylation site ties together numerous regions of the protein through a hydrogen bonding network which includes the activation loop, the R125 form the catalytic loop, and the E/ F linker from the C-lobe, and a His in the C-helix from the N-lobe (see for example / summary Fig 1 of PMID: 22334660). It would be of interest to mention if the AGC-conserved residues that interact with the pT197 (in mammalian PKA) are conserved in PfPKG. If not, this would strongly support the notion that phosphorylation of the T in the activation loop has evolved to fulfill other functions than “conventional” enzyme activation. In this respect, the discussion of similarities with mammalian PKCδ (P. 25) is interesting. Are these “pT-interacting residues” conserved in PKCδ?

**Part III – Minor Issues: Editorial and Data Presentation Modifications**

Reviewer #1: I have only several minor revisions to request.

1. Please check the listing of phosphorylation residues in the text, sometimes the authors write T576 or S576.

2. In the Fourth paragraph of the introduction, please change from “two of these sites are found within CNB-B (T202, Y214) and four within the kinase domain” to “two of these sites are found within CNB-B (T202, Y214) and five within the kinase domain”.

3. Which concentration of Zaprinast was used? Please clarify that in the methods. Is this concentration toxic to the parasite?

4. Clarify why the authors extended the studies to F96.

5. What is the cGMP saturation in PKG sites?

6. Please provide standard deviation (SD) data and apply statistics throughout. Data S1 needs to be improved with units and the number of replicates.

7. The authors state that the binding of PKAtide substrate is not significantly different for WT PfPKG and PfPKGSTmut. It would be beneficial for the manuscript to provide statistical analysis supporting this conclusion. Also, the authors state that despite their attempts, it was not possible to obtain reliable km values for ATP with PfPKGSTmut. Could the authors explain why?

8. In Figure 2H the authors state that there are differences between PfPKGWT and PfPKGSTmut in ML-10 and C2 inhibition values. However, no statistics are available to support the statement. The error bars are very high. Please review and revise as needed.

9. The authors need to improve the quality of Figures 3B and 3C. Figure S5F does not exist.

10. Figures 3F and 3G do not match the textual description of the results. There are differences between DMSO- and RAP-treated parasites at 0.25 uM and 0.5 uM of C2. Please add statistics to corroborate the arguments.

11. Please add values units to all graphs. Also, it would be best to choose one standardized way to present the graph data in the Figures and Supplemental Figures.

12. Have the authors tested C2 and/or ML-10 against Y214A and Y214F mutant lines in order to check whether those mutations (situate away from the ATP binding site) can affect the potency of these compounds (even though the mutations are not in the ATP-binding pocket)? If so please mention.

Reviewer #2: The structure models shown in Figure 1J are cluttered and difficult to interpret. As presented, the dots do not effectively show surfaces/packing, and the various colors of the cartoon portion are somewhat confusing. It is also important to differentiate between actual solved structure (the wild-type) and the two theoretical models that appear to have had the Y214 mutated but no additional modeling (e.g. energy minimisation, which is definitely unnecessary to make the author's point).

Upon introduction of zaprinast it would be helpful to describe what the drug does for the non-specialist audience.

Reviewer #3: In addition to that, the authors might want to consider the following (more minor) points:

P.11, top: “Replacement of Y694 with Ala likely weakens the packing of the kinase activation loop against the α-helix”. Say which alpha-helix, and what the consequences are.

P.11, middle: the fact the Tyr214->Phe substitution in CNB-B has no phenotype (while Tyr214->Ala is lethal) shows indeed that phosphorylation is not needed, but that the residue plays an important structural role. This is compounded by the fact that the corresponding position in CNB-A is a Phe. In Fig 1J, it is not clear how the Ala substitution affects the general environment (it is mentioned that “., and how it is different from the Phe substitution?... the only difference in the panels appears to be the substituted residue itself. Do the authors mean that stacking between F239 and Y214 (in the wild-type) stabilises the overall structure? If so, the mutation may cause lack of binding only due to general structural instability of the CNB. Please clarify (this is addressed in the discussion [P. 23], but could be clarified a bit more. CD analysis of this mutant (as has been done for PKGSTMUT) might be of interest to address the extent of destabilisation.

P.11: monitoring cleavage of SERA5 in quite an indirect proxy for PKG action… Could the PKAtide peptide used in subsequent experiments (Fig 1G) be used to monitor activity, if not in vivo, at least in least from parasites extracts? Also, why use SERA5 as a proxy here, and AMA1 in the experiments in Fig 4F? Please clarify.

P. 15 and Fig 2G: “Attempts however to obtain reliable Km values for ATP with PfPKGSTmut, were unsuccessful, suggestive of subtle structural changes affecting ATP binding but not kinase activity in vitro”. It is mentioned in the Introduction that these sites can also bind cAMP. A wild idea: could it be that the mutations affect ATP binding properties of the CNBs?

P. 18, bottom: the statement “These data suggest that T695 is phosphorylated well before the point of PKG activation at egress. It therefore appears unlikely that T695 phosphorylation contributes to the cGMP-mediated activation mechanism” is a bit misleading. Formally speaking, early phosphorylation might prime the enzyme for subsequent full activation mediated by cGMP binding at a later stage. In another words, this T-loop phosphorylation may be necessary, but not sufficient, for activation. The rest of the data show this is not the case, but this particular statement is logically flawed.

P.20: “The increase in hyperphosphorylated sites observed in both T695A

and T695E phosphoproteomes raised the question of whether these mutations could

alter the substrate specificity of PfPKG”. Actually, the increase in hypophosphorylated sites raise the exact same question! :-) In the view of this reviewer, any change (increase/decrease/hypo/hyper) actually does.

Discussion: “The exact mechanism by which Y214A parasites overcome the lethal phenotype, in the presence of zaprinast, is not clear.” Could it be that the structural changes caused by the mutation have a direct effect (that does not require “release of of the kinase domain”)?

P.21, substrate preference: “We observed for both mutants a more relaxed specificity for Thr in positions -1 to +4” . Interestingly, the phosphopeptide corresponding to T695

(AYpTLVGTPHYMAPEVILGK) actually complies with that (T at +4). Could there be any significance to that?!

Discussion: “Future studies should explore whether activation loop phosphorylation is

an autophosphorylation event or whether it is mediated by PDK1”. Please mention which kinase is the

PLOS authors have the option to publish the peer review history of their article (what does this mean?). If published, this will include your full peer review and any attached files.

Reviewer #1: No

Reviewer #2: No

Reviewer #3: **Yes: **Christian Doerig

Figure Files:

While revising your submission, please upload your figure files to the Preflight Analysis and Conversion Engine (PACE) digital diagnostic tool, https://pacev2.apexcovantage.com. PACE helps ensure that figures meet PLOS requirements. To use PACE, you must first register as a user. Then, login and navigate to the UPLOAD tab, where you will find detailed instructions on how to use the tool. If you

---

## [Editor Report · Decision Letter 1]

20 Jun 2024

Dear Dr. Koussis,

We are pleased to inform you that your manuscript 'Activation loop phosphorylation and cGMP saturation of PKG regulate egress of malaria parasites.' has been provisionally accepted for publication in PLOS Pathogens.

Best regards,

Vasant Muralidharan

Guest Editor

PLOS Pathogens

Margaret Phillips

Section Editor

PLOS Pathogens

Michael Malim

Editor-in-Chief

PLOS Pathogens

orcid.org/0000-0002-7699-2064
---

## [Editor Report · Acceptance letter]

24 Jun 2024

Dear Dr. Koussis,

We are delighted to inform you that your manuscript, "Activation loop phosphorylation and cGMP saturation of PKG regulate egress of malaria parasites.," has been formally accepted for publication in PLOS Pathogens.

Best regards,

Michael Malim

Editor-in-Chief

PLOS Pathogens

orcid.org/0000-0002-7699-2064